# 🤖 E.T. Bench: Towards Open-Ended Event-Level Video-Language Understanding

Ye Liu[1,2], Zongyang Ma[2,3], Zhongang Qi[4*†], Yang Wu[5], Ying Shan[2], Chang Wen Chen[1†]

[1] The Hong Kong Polytechnic University  [2] ARC Lab, Tencent PCG
[3] Chinese Academy of Sciences  [4] Huawei Noah's Ark Lab  [5] Tencent AI Lab
coco.ye.liu@connect.polyu.hk
https://polyu-chenlab.github.io/etbench/

## Abstract

Recent advances in Video Large Language Models (Video-LLMs) have demonstrated their great potential in general-purpose video understanding. To verify the significance of these models, a number of benchmarks have been proposed to diagnose their capabilities in different scenarios. However, existing benchmarks merely evaluate models through video-level question-answering, lacking fine-grained event-level assessment and task diversity. To fill this gap, we introduce **E.T. Bench** (**E**vent-Level & **T**ime-Sensitive Video Understanding **Bench**mark), a large-scale and high-quality benchmark for open-ended event-level video understanding. Categorized within a 3-level task taxonomy, E.T. Bench encompasses 7.3K samples under 12 tasks with 7K videos (251.4h total length) under 8 domains, providing comprehensive evaluations. We extensively evaluated 8 Image-LLMs and 12 Video-LLMs on our benchmark, and the results reveal that state-of-the-art models for coarse-level (video-level) understanding struggle to solve our fine-grained tasks, *e.g.*, grounding event-of-interests within videos, largely due to the short video context length, improper time representations, and lack of multi-event training data. Focusing on these issues, we further propose a strong baseline model, **E.T. Chat**, together with an instruction-tuning dataset **E.T. Instruct 164K** tailored for fine-grained event-level understanding. Our simple but effective solution demonstrates superior performance in multiple scenarios.

## 1 Introduction

The recent advent of Multi-modal Large Language Models (MLLMs) [2, 91, 59, 115, 66, 17] has led to a substantial paradigm shift in visual-language understanding, moving away from designing task-specific models and collecting domain-specific data, towards developing general-purpose task-solvers for open-ended scenarios. By integrating LLMs with visual encoders, these models jointly benefit from perception abilities and powerful reasoning skills, showing remarkable performance in even unseened applications, demonstrating great potential for such scheme.

To effectively evaluate the capabilities of these models, a number of benchmarks [107, 45, 62, 14, 106, 48, 69, 50, 63] have been introduced to study their feasibilities in different scenarios. Yet, most of the benchmarks are focusing on image or short (seconds-long) video understanding, which require strong static scene understanding abilities but overlooking the fine-grained temporal information. Some recent works [86, 67] tend to evaluate MLLMs on longer videos, but they still leverage multiple-choice question-answering (MCQ) as their main task, lacking flexibilities for open-ended tasks. Nevertheless, none of the existing benchmarks are designed for multi-event or time-sensitive scenarios, thus they

---

*Work done at ARC Lab, Tencent PCG    †Corresponding authors

38th Conference on Neural Information Processing Systems (NeurIPS 2024) Track on Datasets and Benchmarks.

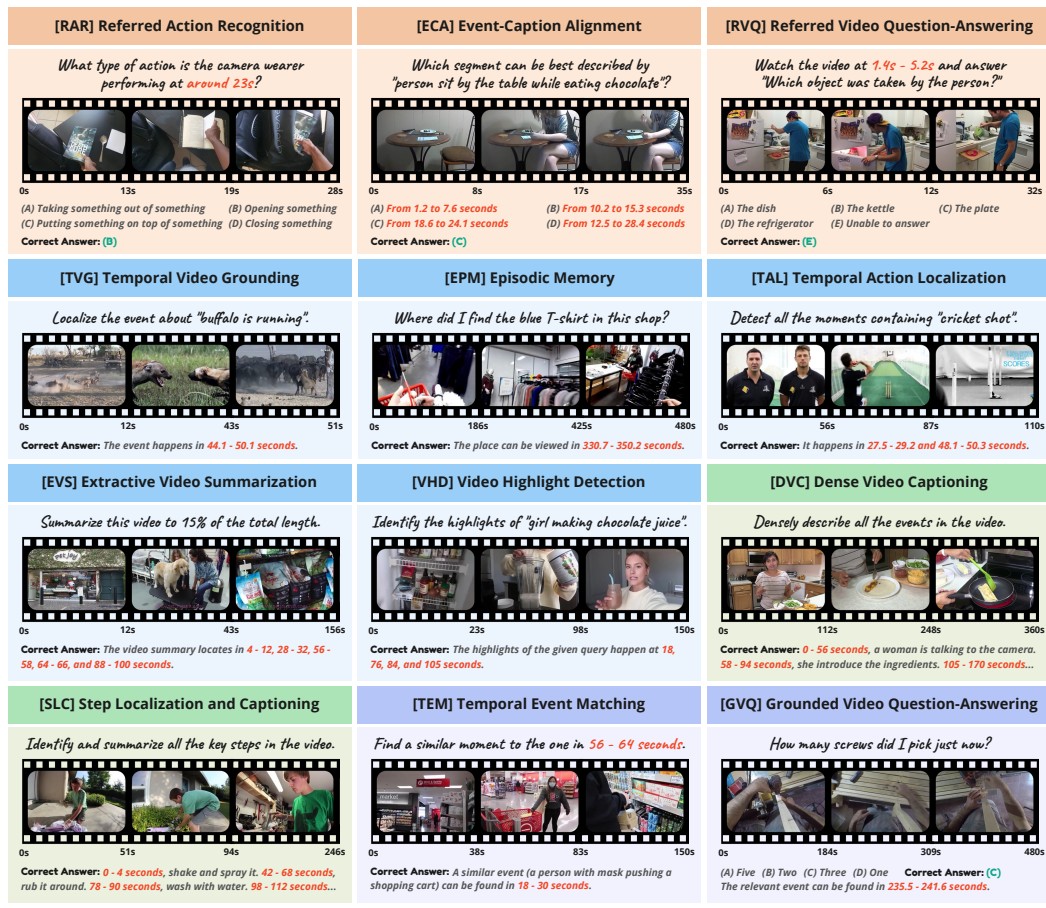

Figure 1: **Task definitions in E.T. Bench.** The 12 tasks derives from 4 essential capabilities for time-sensitive video understanding: *referring*, *grounding*, *dense captioning*, and *complex understanding*.

suffer from severe single-frame biases, as can be seen in the comparable performances between Image- and Video-LLMs in these benchmarks [45, 48].

To address these issues and better understand the open-ended capabilities of these models, we propose **E.T. Bench**, a comprehensive benchmark for event-level and time-sensitive video understanding. As shown in Figure 1 and compared in Table 1, our benchmark significantly diverse from previous ones that it focus on time-sensitive understanding on long and multi-event videos. Our motivation is that a well-performing Video-LLM should possess the capability to precisely refer to and localize any events that align with user interests. Based on this assumption, we build our task taxonomy by summarizing four essential capabilities required for time-sensitive video understanding: *referring*, *grounding*, *dense captioning*, and *complex understanding*. Then, each capability is delineated with carefully designed tasks. The diversity of scenarios is ensured by meticulously collecting videos from 15 datasets covering 8 domains. A comprehensive data cleaning, annotation repurposing, instruction design, manual verification, and sampling pipeline is leveraged to generate 7.8K high-quality annotations.

We extensively evaluate 20 models, including 7 open-source Image-LLMs, 9 open-source Video-LLMs, and 4 commercial MLLMs, on E.T. Bench. The results reveal that state-of-the-art models on existing VideoQA benchmarks [45, 69, 48] struggle on our E.T. Bench, especially on *grounding*, *dense captioning*, and *complex understanding* tasks. We attribute it to two key limitations of existing development pipelines for MLLMs. First, the discrete next-token prediction paradigm has natural drawbacks in numerical calculations [27, 38], limiting timestamp understanding and generation. Second, most existing video instruction-tuning datasets [47, 66, 49] predominantly comprise short videos with coarse-level annotations, brining significant gap between training and real-world applications. To tackle these problems, we propose **E.T. Chat**, a novel time-sensitive Video-LLM that reformulates timestamp prediction as an embedding matching problem, serving as a

Table 1: Quantitative comparison between E.T. Bench and existing Video-LLM benchmarks.

| Benchmark | Domain | Annotator | #Tasks | #Samples | #Videos | Avg./Max. Duration | Long Video | Event Level | Time Sensitive | Answer Type | Evaluation Method |
|---|---|---|---|---|---|---|---|---|---|---|---|
| SEED-Bench [45] | Action | LLM | 3 | 3,757 | 3,757 | 8 frames | ✗ | ✗ | ✗ | MCQ | Likelihood |
| EgoSchema [67] | Egocentric | LLM | 1 | 5,031 | 5,031 | 180s / 180s | ✓ | ✗ | ✗ | MCQ | Likelihood |
| AutoEval-Video [16] | Open | Human | 9 | 327 | 327 | 15s / 101s | ✗ | ✗ | ✗ | Open | GPT-4 |
| Video-Bench [69] | Open | LLM | 1 | 17,054 | 5,917 | 56s / 3,599s | ✓ | ✗ | ✗ | MCQ | Mixed‡ |
| TempCompass [63] | Open | LLM | 4 | 7,540 | 410 | 12s / 35s | ✗ | ✗ | ✗ | MCQ/Open | Rule/GPT |
| MVBench [48] | Open | Human† | 1 | 4,000 | 3,673 | 15s / 116s | ✗ | ✗ | ✗ | MCQ | Rule |
| **E.T. Bench** (Ours) | Open | Human† | 12 | 7,289 | 7,002 | 129s / 795s | ✓ | ✓ | ✓ | MCQ/Open | Rule |

† Repurposed from existing datasets    ‡ Including next-token likelihood, T5 sentence similarity, and GPT-3.5 assisted evaluation

strong baseline on E.T. Bench. As for data, we also curate **E.T. Instruct 164K**, an instruction-tuning dataset tailored for multi-event and time-sensitive scenarios. Extensive comparisons demonstrate the effectiveness of the proposed model and dataset. We hope that the proposed benchmark, model, and dataset can inspire future research on Video-LLMs.

## 2 Event-Level & Time-Sensitive Video Understanding Benchmark

In this section, we illustrate the detailed pipeline employed to develop our E.T. Bench. As shown in Figure 2 (right), the pipeline begins with the definition of four essential capabilities for event-level and time-sensitive video understanding, *i.e.*, *referring*, *grounding*, *dense captioning*, and *complex understanding*, arranged in increasing order of difficulty. For each capability, we design a series of tasks specifically for effective assessment of the respective capability. For each task, we meticulously select existing datasets with timestamp annotations provided by human annotators, and rewrite them into instruction-following formats according to task formulation, ensuring high quality and verisimilitude. The diversity of E.T. Bench is guaranteed by carefully choosing variable-length videos from different domains. Finally, a thorough manual check, filtering, and sampling process is conducted to eliminate unsatisfactory samples. Details for each step are introduced as follows.

### 2.1 Hierarchical Task Taxonomy

To evaluate the open-ended Video-LLMs from various perspectives, we design a three-level task taxonomy depicted in Figure 2 (left). Definitions for capabilities and tasks are as follows.

**Referring** means the ability to comprehend time information from user inputs. For example, given a question *"What is the person doing from 23s to 35s?"*, the model has to understand which part of the video is the user referring to, and provide response with more consideration on that segment. For better quantifying the model performances, we formulate all the *referring* tasks as multiple-choice question-answerings (MCQs), including 1) [RAR] *Referred Action Recognition*: Identify the action given a coarse timestamp hint (*e.g.*, *"around 12s"*). The model has to determine the actual reference according to both the video and options. 2) [ECA] *Event-Caption Alignment*: Select the correct temporal boundary for a given caption. The model need to understand and distinguish multiple timestamps in the options. 3) [RVQ] *Referred Video Question-Answering*: Answer the question conditioning on a given segment. Each question is supplied with four candidates and an "unable to answer" option, denoting the case when it cannot be answered with given the segment.

**Grounding** indicates the ability to localize event- or moment-of-interests with accurate timestamps. It diverse from previous works [48, 101] that only consider coarse-level grounding, *e.g.*, *"at the beginning/middle/end of the video"*. Outputs for *grounding* tasks are open-ended, processed by rule-based parsers and evaluated with continuous metrics. The definitions of tasks are 1) [TVG] *Temporal Video Grounding*: Determine the temporal boundary of a single event according to the text description. 2) [EPM] *Episodic Memory*: Localize the event that can answer the given question in egocentric scenarios, *e.g.*, *"Where is my backpack?"*. 3) [TAL] *Temporal Action Localization*: Detect and localize a series of segments containing the given action, *e.g.*, finding all "golf swing" segments in a long video. 4) [EVS] *Extractive Video Summarization*: Provide a list of segments that can be merged to form a compact video summary (with around 15% of the total duration). 5) [VHD] *Video Highlight Detection*: Cherry-pick a single timestamp (*e.g.*, *"15s"*) that can best reflect the highlight

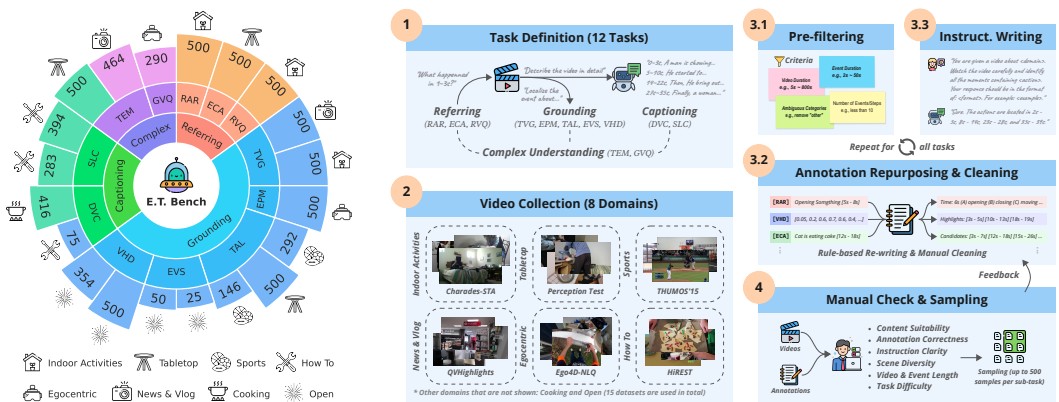

Figure 2: **Left:** Task taxonomy and sample distribution. **Right:** Generation pipeline for E.T. Bench. We conduct a thorough process of pre-filtering, annotation repurposing, instruction writing, manual check, and sampling to obtain high-quality fine-grained annotations. Details discussed in Section 2.

moment corresponding to a query. Note that the formulations of [TAL], [EVS], and [VHD] have been modified (compared with previous works [83, 87, 44]) to fit the nature of LLMs.

**Dense Captioning** is a more complicated ability that requires jointly localize key-events and generate descriptions/summaries for each segment. This is more practical for storytelling or key-information extraction from long videos in comparison with video-level captioning on trimmed clips [103, 12]. We define two *dense captioning* tasks according to different goals: 1) [DVC] *Dense Video Captioning*: Comprehensively describe all the events happened in the video. This is a general case with the goal of covering as much events as possible. 2) [SLC] *Step Localization and Captioning*: Identify and describe only the key-steps in instructional videos. In this case, the segments are shorter & disjoint and the step descriptions are more precise compared with [DVC].

**Complex Understanding** refers to the versatile integration of the aforementioned time comprehension and event localization, requiring the model to demonstrate proficient event-level and time-sensitive understanding. The two tasks are: 1) [TEM] *Temporal Event Matching*: Find and locate a similar event in the same video conditioning on the given segment. This involves a two-stage reasoning process that first identify the event in the given timestamps, then localize another segment with the most similar content. 2) [GVQ] *Grounded Video Question-Answering*: Answer the given multiple-choice question by selecting an option and localizing a segment that supports the answer. This is also a complex scenario requiring both understanding and localization abilities. Validating the localization results can help diagnose the reasoning process of Video-LLMs.

## 2.2 Data Collection and Annotation

The key challenge of data collection is how to obtain videos with precise temporal boundary annotations. Previous works either prompt LLMs with frame-level information extracted from collections of experts [100, 76, 101, 34] or transform human-annotated moment tags (*e.g.*, 5.2s) to boundaries (*e.g.*, 3.2s to 7.2s) using pre-defined rules [56, 78]. These solutions can only generate temporal boundaries with substantial noise, which are not suitable for accurate model evaluation. Therefore, we meticulously curate multi-event videos from existing datasets with high-quality human-annotated timestamps, and repurpose the annotations by transforming them according to our task formulations. To ensure the diversity of scenarios, we carefully select 15 datasets from 8 domains, *i.e.*, *indoor activities*, *tabletop*, *sports*, *egocentric*, *cooking*, *news & vlogs*, *how-to*, and *open*. All the videos are collected from `val` or `test` splits to prevent potential data leakage. For annotation generation, we develop for each task a thorough process containing pre-filtering, manual annotation repurposing, and instruction template design, presented in Figure 2 (right). Please refer to Section A for detailed task-specific pre-filtering criteria, repurposing methods, and instruction templates.

After annotation generation, we conduct a careful manual review on the samples, focusing on content suitability, annotation correctness, instruction clarity, scene diversity, video & event length, and task difficulty. Feedback from this review helped us actively optimize the generation process. Finally, to

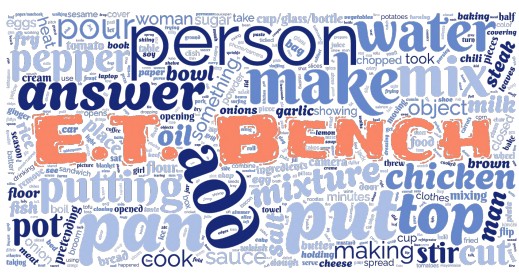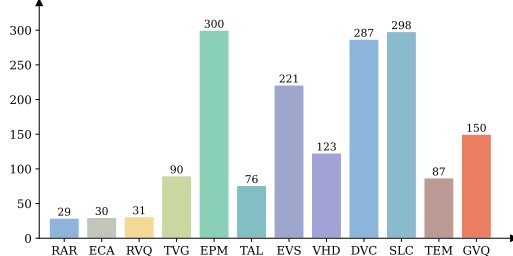

Figure 3: **Left:** Word cloud of text queries shows a considerable degree of diversity. **Right:** Distribution of averaged video durations (in seconds) across 12 tasks.

balance the quality and efficiency of evaluation, we randomly sample up to 500 samples for each sub-task (task-source combination). In most cases, each video would only be sampled once.

## 2.3 Benchmark Analysis

We present some statistics of the generated benchmark. Detailed comparisons are shown in Table 1. Overall, the proposed E.T. Bench contains 7,289 samples under a taxonomy of 4 capabilities, 12 tasks, and 20 sub-tasks. There are in total 7,002 unique videos originate from 15 datasets, covering 8 domains. The average duration of videos is 129.3s, with the minimum and maximum values of 6.2s and 795.0s, respectively. This differs from most existing benchmarks that have averaged durations only $10 \sim 20$ seconds. We also have diverse answer types for different tasks, including both MCQ and open-ended styles. The evaluation process is purely rule-based without human or LLM integration, ensuring satisfied objectivity. Below we introduce more detailed analysis on the benchmark.

**Task and Sample Distribution.** Figure 2 (left) shows the distribution of tasks, sub-tasks, and samples in E.T. Bench. Here, a sub-task is defined as a task-source combination, *e.g.*, [TVG] contains two sub-tasks from two source datasets. A large proportion of samples are in the *grounding* category, as we emphasize the moment localization ability of modern Video-LLMs.

**Text Queries.** Figure 3 (left) shows the word cloud of text queries in E.T. Bench. Thanks to the wide range of video domains, the queries are also diverse in terms of both nouns and verbs. Most queries are human-centric, describing human activities or human-object interactions. Please refer to Section A.4 for distributions of nouns and verbs.

**Video Durations.** Figure 3 (right) shows the distribution of averaged video durations across tasks. Our videos have a wide spectrum of durations, where *referring* tasks have relatively shorter videos, and *grounding* and *captioning* have longer ones. Our experimental results in Table 3 show that the duration of videos have significant influence on model performance.

## 3 Our Method

Extensive evaluations (in Table 3) reveal that even the state-of-the-art Video-LLMs cannot perform well on E.T. Bench, especially on the more complicated *grounding*, *dense captioning*, and *complex understanding* tasks. We attribute this phenomenon to two key limitations of existing development pipelines for Video-LLMs: 1) **Model**: Existing models fall short in numerical modeling [21, 23], which are essential capabilities for arithmetic calculations – timestamps processing in our case; 2) **Data**: Both pre-training and instruction-tuning are conducted on short & single-event videos, leading to weak general understanding abilities for multi-event videos. To address these limitations, we propose **E.T. Chat**, a novel Video-LLM that reformulates timestamp prediction as an embedding matching problem, serving as a strong baseline on E.T. Bench. we also curate **E.T. Instruct 164K**, an instruction-tuning dataset tailored for multi-event and time-sensitive video understanding.

### 3.1 Model

Figure 4 presents the overall architecture of E.T. Chat. Given a video frame $\mathbf{V}_t \in \mathbb{R}^{H \times W \times 3}$ sampled at time $t \in T$, where $H$ and $W$ are the height and width, we first leverage a frozen visual encoder

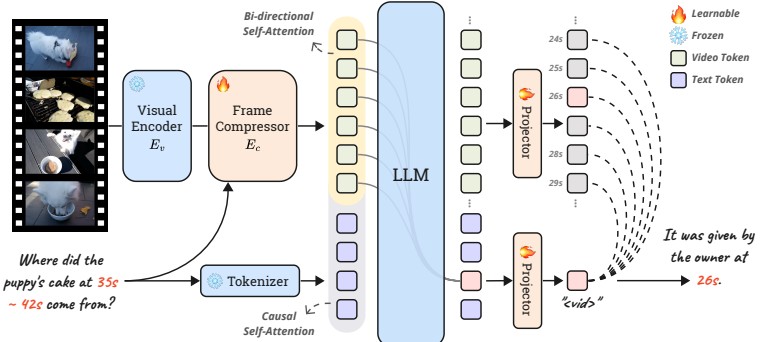

Figure 4: **Overall architecture of E.T. Chat.** We reformulate timestamp prediction as an embedding matching problem. See Section 3 for details.

Table 2: Distribution of E.T. Instruct 164K.

| Source | #Samples |
|---|---|
| HowToCaption [84] | 16,907 |
| DiDeMo [5] | 33,000 |
| QueryD [70] | 4,267 |
| TACoS [79] | 6,693 |
| ActivityNet [10] | 19,637 |
| HACS [112] | 15,218 |
| NaQ [78] | 10,546 |
| VideoXum [55] | 7,989 |
| Mr. HiSum [88] | 9,056 |
| ViTT [35] | 2908 |
| COIN [90] | 7,659 |
| HowToStep [52] | 20,000 |
| EgoTimeQA [20] | 10,000 |
| Total | 163,880 |

$E_v$ to convert it into patch embeddings $\mathbf{P}_t \in \mathbb{R}^{K \times C}$, where $K$ and $C$ are the number of patches and feature dimension. To preserve high temporal resolution while reducing redundant compute, we adopt a frame compressor $E_c$ to merge and project patch embeddings to a single token $\mathbf{e}_v^t \in \mathbb{R}^{1 \times D}$, where $D$ is the embedding dimension of LLM. The compressed frame tokens $\{\mathbf{e}_v^t\}_{t=1}^T$ are then concatenated with text tokens $\{\mathbf{e}_q^n\}_{n=1}^N$ and sent into LLM for response generation.

**Frame Compression.** As illustrated in Figure 5, the frame compressor $E_c$ arises from [51] and consists of a Q-Former [46] $E_q$ with $M$ learnable queries, a context aggregator $E_a$, and a projector $E_p$. For each time step, $E_q$ accepts patch embeddings $\mathbf{P}_t$ and the text prompt $\mathbf{T}$ as inputs, and resamples them into learnable queries $\mathbf{Q}_t \in \mathbb{R}^{M \times C}$. Then, $E_a$ merges $\mathbf{Q}_t$ with $\mathbf{P}_t$ and compresses them into a single token. $E_p$ finally projects it to the same embedding space as LLM. In particular, the context aggregator $E_a$ is built upon cross attention module [96], formulated as follows:

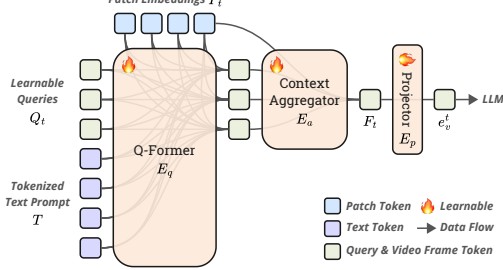

Figure 5: **Detailed illustration of frame compressor.** It accepts video patch embeddings $\mathbf{P}_t$ and the text prompt $\mathbf{T}$ as inputs, and compress video frame features into a single token.

$$\mathbf{a} = \text{Softmax}\left(\frac{(\mathbf{w}_q \times \mathbf{Q}_t)^\top \times (\mathbf{w}_k \times \mathbf{P}_t)}{\sqrt{C}}\right) \quad (1)$$

$$\mathbf{F}_t = \text{Mean}(\mathbf{a} \times \mathbf{P}_t + \mathbf{Q}_t) \quad (2)$$

Here, $\mathbf{F}_t \in \mathbb{R}^{1 \times C}$ is the compressed frame token for time $t$, containing text-conditioned visual information. This process can be parallelized across all frames. The projected video token sequence $\mathbf{e}_v$ is then concatenated with text tokens $\mathbf{e}_q$ to form the inputs with shape $(T + N) \times D$ for LLM.

**Timestamp Prediction as Embedding Matching.** Our key insight focuses on the design of timestamp processing. As discussed in Section B.1, we claim that directly generating continuous signals (*i.e.*, timestamps in our case) via discrete next-token prediction is sub-optimal. Motivated by the characteristic of Transformers that they are naturally good at selective copying rather than numerical calculations [27, 38], we propose to reformulate timestamp prediction as an *embedding matching* problem. That is, we train the model to generate/copy embeddings of video frames that it would like to refer to, and obtain timestamps by matching these embeddings back to the video.

Specifically, we define a special token `<vid>` used to stimulate the matching process. When `<vid>` is generated during inference, *e.g.*, the model outputs *"the event happens around `<vid>`"*, this token is utilized to match a video frame token, such that the desired timestamp can be easily obtained from the matched frame index. For example, for a video sampled to 1 FPS, if `<vid>` is matched to the $i$-th frame, then `<vid>` means the $i$-th second of the video. The matching process is designed to be simple and efficient. We denote the $l$-th layer hidden states of `<vid>` token and video frame tokens as $\mathbf{h}_{vid}^l \in \mathbb{R}^{1 \times D}$ and $\mathbf{h}_{frm}^l \in \mathbb{R}^{T \times D}$, respectively. During matching, two MLPs $E_{vid}$ and $E_{frm}$ are first leveraged to project the hidden states to the alignment space $g$:

$$\mathbf{g}_{vid} = E_{vid}(\mathbf{h}_{vid}^{L-1}), \quad \mathbf{g}_{frm} = E_{frm}(\mathbf{h}_v^L) \quad (3)$$

Here, $L$ refers to the total number of LLM layers. We extract `<vid>`'s hidden states from the second-last layer to preserve a larger feature range [30]. Subsequently, we compute the cosine similarities between $\mathbf{g}_{vid}$ and all $\{\mathbf{g}_{frm}^t\}_{t=1}^T$ to obtain the matched frame index $t_{match}$:

$$\mathbf{s} = \frac{\mathbf{g}_{vid} \cdot \mathbf{g}_{frm}}{||\mathbf{g}_{vid}||_2 \cdot ||\mathbf{g}_{frm}||_2} \in \mathbb{R}^{1 \times T}, \quad t_{match} = \mathrm{argmax}(\mathbf{s}) \tag{4}$$

The frame index $t_{match}$ is then multiplied with frame rate $r$ to generate the real timestamp in seconds. Through this operation, the direct prediction of timestamps is replaced by embedding matching, which is easier to learn as a selective copying problem by Transformer-based models. For the case when `<vid>` occurs in inputs, the input features of `<vid>` are added with the corresponding frame features. During training, an extra binary matching loss is utilized:

$$\mathcal{L}_{matching} = -\frac{1}{T} \sum_{t=1}^T \mathbf{y}_t \cdot \log(\mathbf{s}_t) \tag{5}$$

Here, $\mathbf{y}_t$ denotes the binary label indicating whether $t$ is the ground truth frame. $\mathcal{L}_{matching}$ is added with the original language modeling loss $\mathcal{L}_{language}$ to jointly optimize the model.

**Numerical Continuity.** The matching process above still cannot preserve numerical continuities among them, as the hidden states of adjacent frames may be far away from each other. We introduce two modifications to effectively alleviate this problem. First, we observe that the causal self-attentions in LLM block out the bi-directional information flow. This is reasonable for text but limits the ability of video understanding. Thus, we allow bi-directional attentions among video tokens. Second, we introduce a smoothed label $\frac{1}{\alpha^{|t-t_{gt}|}}$ to replace the binary label $\mathbf{y}_t$ in Eq. 5, where $\alpha$ is a hyper-parameter controlling the extent of smoothing, and $t_{gt}$ refers to the ground truth frame index.

### 3.2   Instruction-Tuning Dataset

The proposed E.T. Instruct 164K contains multi-event understanding samples generated from 14 source datasets, illustrated in Table 2. It covers a wide range of event-level understanding tasks, including temporal grounding, summarization, highlight detection, dense captioning, and question-answering. More details about the generation process are presented in Section C.

## 4   Experiments

### 4.1   Evaluation Settings

As different tasks in E.T. Bench are under different settings with diverse output formats. A single metric (*e.g.*, accuracy) like existing benchmarks is not sufficient. To balance the quantity of metrics and the ease of ranking, we unify the metrics within each capability and leverage accuracy for *referring* tasks, F1 score for *grounding* tasks, F1 score and sentence similarity for *dense captioning* tasks, and recall for *complex understanding* tasks. Detailed metrics are introduced in Section D.1.

### 4.2   Main Results

We extensively evaluate 7 open-source Image-LLMs, 9 open-source Video-LLMs, and 4 commercial MLLMs on E.T. Bench. Details of each model are introduced in Section D.2. For Image-LLMs, we uniformly sample 8 frames and add an extra prompt indicating the video duration as a hint for timestamps. For Video-LLMs, we use their default number of frames as inputs. Commercial MLLMs are evaluated by calling APIs on a subset with 470 samples. The inputs for GPT-4V and GPT-4o are aligned with Image-LLMs. For Gemini-1.5 models, raw videos are directly uploaded.

The evaluation results are presented in Table 3. We report the metrics averaged among sub-tasks due to space limit. The *Random* in the first row refers to random guessing. We also provide comparisons and ranking in Figure 6. Below we summarize our key findings from the results.

**Performance gap between Image- and Video-LLMs.** We observe that on *referring* tasks, most Image- and Video-LLMs perform at the same level. Some Image-LLMs such as XComposer and Qwen-VL-Chat can even beat most Video-LLMs. This is because the videos for *referring* are generally short, as compared in Figure 3 (right), such that the sampled 8 frames cover most information. The gap

Table 3: **Performance of representative MLLMs on E.T. Bench.** We evaluate both open-source and commercial models. The best and second-best results are marked purple and orange, respectively.

| Method | Referring | | | Grounding | | | | | Dense Captioning | | | | Complex | |
|---|---|---|---|---|---|---|---|---|---|---|---|---|---|---|
| | $RAR_{Acc}$ | $EVC_{Acc}$ | $RVQ_{Acc}$ | $TVG_{F1}$ | $EPM_{F1}$ | $TAL_{F1}$ | $EVS_{F1}$ | $VHD_{F1}$ | $DVC_{F1}$ | $DVC_{Sim}$ | $SLC_{F1}$ | $SLC_{Sim}$ | $TEM_{Rec}$ | $GVQ_{Rec}$ |
| *Random* | 25.0 | 25.0 | 20.0 | – | – | – | – | – | – | – | – | – | – | – |
| *Open-source Image-LLMs: All models use 8 uniformly sampled frames as inputs. Prompts have been added with hints about timestamps.* | | | | | | | | | | | | | | |
| LLaVA-1.5 [58] | 34.2 | 27.4 | 26.2 | 6.1 | 1.9 | 7.8 | 2.4 | 30.9 | 14.5 | 11.5 | 0.9 | 9.5 | 7.7 | 0.0 |
| LLaVA-InternLM2 [11] | 34.0 | 34.8 | 37.0 | 2.7 | 0.1 | 0.3 | 0.2 | 32.3 | 16.9 | 8.5 | 0.1 | 4.7 | 7.2 | 1.5 |
| mPLUG-Owl2 [105] | 37.8 | 26.4 | 34.6 | 1.1 | 0.2 | 3.0 | 4.1 | 36.8 | 0.1 | 8.1 | 0.1 | 7.7 | 6.2 | 0.0 |
| XComposer [111] | 33.0 | 19.6 | 40.2 | 4.9 | 1.5 | 9.9 | 2.8 | 28.9 | 5.4 | 5.9 | 2.7 | 9.0 | 10.5 | 0.0 |
| Bunny-Llama3-V [31] | 33.2 | 27.4 | 26.6 | 7.0 | 0.1 | 5.1 | 0.4 | 30.6 | 13.5 | 8.8 | 0.1 | 7.6 | 7.2 | 0.0 |
| MiniCPM-V-2.5 [93] | 37.6 | 28.0 | 37.6 | 2.0 | 0.1 | 4.4 | 13.4 | 18.7 | 6.2 | 11.8 | 1.4 | 9.7 | 0.7 | 0.0 |
| Qwen-VL-Chat [7] | 33.4 | 32.2 | 33.6 | 16.2 | 4.0 | 10.7 | 16.3 | 34.4 | 17.4 | 13.8 | 6.2 | 13.1 | 3.2 | 1.5 |
| *Open-source Video-LLMs: All models use their default numbers of frames as inputs.* | | | | | | | | | | | | | | |
| Video-ChatGPT [66] | 22.6 | 24.2 | 23.0 | 7.0 | 1.3 | 15.1 | 8.4 | 28.8 | 8.8 | 11.3 | 5.7 | 10.2 | 15.9 | 0.0 |
| Video-LLaVA [53] | 33.6 | 33.0 | 22.6 | 7.0 | 1.9 | 15.0 | 0.3 | 28.9 | 28.0 | 15.0 | 0.9 | 8.3 | 7.5 | 0.1 |
| LLaMA-VID [51] | 30.4 | 38.4 | 28.8 | 5.5 | 1.2 | 8.0 | 1.4 | 30.0 | 27.1 | 12.6 | 5.2 | 11.1 | 7.0 | 0.9 |
| Video-LLaMA-2 [110] | 28.8 | 27.4 | 28.0 | 0.1 | 0.0 | 0.0 | 0.0 | 1.5 | 0.6 | 14.5 | 0.0 | 15.2 | 0.0 | 0.1 |
| PLLaVA [104] | 33.8 | 22.6 | 31.8 | 6.9 | 1.1 | 5.7 | 0.3 | 28.9 | 13.3 | 10.6 | 9.7 | 11.8 | 4.1 | 1.2 |
| VTimeLLM [33] | 28.4 | 31.0 | 29.2 | 7.6 | 1.9 | 18.2 | 15.9 | 28.9 | 12.4 | 13.1 | 8.7 | 6.4 | 6.8 | 1.9 |
| VTG-LLM [28] | 6.6 | 12.0 | 7.8 | 15.9 | 3.7 | 14.4 | 26.8 | 48.2 | 40.2 | 18.6 | 20.8 | 14.4 | 8.9 | 1.4 |
| TimeChat [82] | 30.8 | 27.6 | 24.6 | 26.2 | 3.9 | 10.1 | 29.1 | 40.5 | 16.6 | 12.5 | 5.6 | 9.2 | 18.0 | 1.5 |
| LITA [34] | 33.0 | 40.8 | 27.2 | 22.2 | 4.6 | 18.0 | 29.7 | 23.9 | 39.7 | 17.2 | 21.0 | 12.2 | 16.0 | 2.2 |
| **E.T. Chat** (Ours) | 44.6 | 37.0 | 33.6 | 38.6 | 10.2 | 30.8 | 25.4 | 62.5 | 38.4 | 19.7 | 24.4 | 14.6 | 16.5 | 3.7 |
| *Commercial MLLMs: Evaluated on a subset with 470 samples.* | | | | | | | | | | | | | | |
| GPT-4V [71] | 33.3 | 40.9 | 46.2 | 27.0 | 1.8 | 18.0 | 28.6 | 55.1 | 16.1 | 19.4 | 21.9 | 13.5 | 23.9 | 0.0 |
| GPT-4o [72] | 27.8 | 27.3 | 57.7 | 40.4 | 4.5 | 20.0 | 17.6 | 56.9 | 46.9 | 22.3 | 23.1 | 14.9 | 13.6 | 0.0 |
| Gemini-1.5-Flash [80] | 38.9 | 50.0 | 61.5 | 43.9 | 5.4 | 27.0 | 5.4 | 60.8 | 31.6 | 14.9 | 16.5 | 13.3 | 20.8 | 1.0 |
| Gemini-1.5-Pro [80] | 61.1 | 27.3 | 57.7 | 43.1 | 6.2 | 33.8 | 7.9 | 47.0 | 24.0 | 17.5 | 5.8 | 9.8 | 32.1 | 1.0 |
| **E.T. Chat**† (Ours) | 33.3 | 31.8 | 30.8 | 32.9 | 8.9 | 28.1 | 15.3 | 60.9 | 39.8 | 19.5 | 23.7 | 14.8 | 14.8 | 1.3 |

† Evaluated on the same subset as commercial MLLMs

becomes larger on tasks with longer videos such as [TVG] and [TAL], demonstrating the importance of temporal modeling on E.T. Bench compared with other benchmarks.

**Strong Video-LLMs on existing benchmarks struggle on E.T. Bench.** Some state-of-the-art Video-LLMs on existing benchmarks, *e.g.*, Video-LLaMA-2 and PLLaVA, are less effective on our E.T. Bench, especially on *grounding* and *dense captioning* tasks. We attribute this to the single-frame bias caused by both model architecture and training data. It also motivate us to consider the balance between spatial and temporal modeling in Video-LLMs.

**Some Video-LLMs fail to follow instructions.** During evaluation, we also notice that some models, *e.g.*, Video-LLaVA and Video-LLaMA-2, fail to generate outputs in desired formats for some tasks even with carefully designed instructions and examples. For instance, Video-LLaVA can only generate repeated text outputs without any timestamps for [EVS], while Video-LLaMA-2 faces the similar problem on almost all *grounding* tasks. we claim that this is due to the severe overfitting on their instruction-tuning data, which do not contain any timestamps outputs.

**Some Image-LLMs performs exceptionally well.** Qwen-VL-Chat and Bunny-Llama-3-8B-V are two models producing relatively good results compared with other Image-LLMs and even some Video-LLMs. On [TVG] and [DVC], Qwen-VL-Chat achieves 15.6 and 21.3 F1 scores, respectively, surpassing a number of Video-LLMs. But there is still a significant gap when compared with best-performing Video-LLMs such as TimeChat and LITA.

**Time-sensitive Video-LLMs are the first-class models.** VTimeLLM, TimeChat, and LITA are three Video-LLMs with explicit optimizations for timestamps modeling, such that they can persistently follow instructions and generate considerable responses.

**Commercial MLLMs are still competitive.** Even with 8-frame inputs, GPT-4V and GPT-4o show their significance compared with open-source models on some tasks such as [TVG], [TAL], and [DVC]. By supporting direct video inputs, Gemini-1.5 series achieve the strongest performance on a number of tasks in E.T. Bench, including [RAR], [EVC], [RVQ], [TVG], [TAL], [VHD], and [TEM].

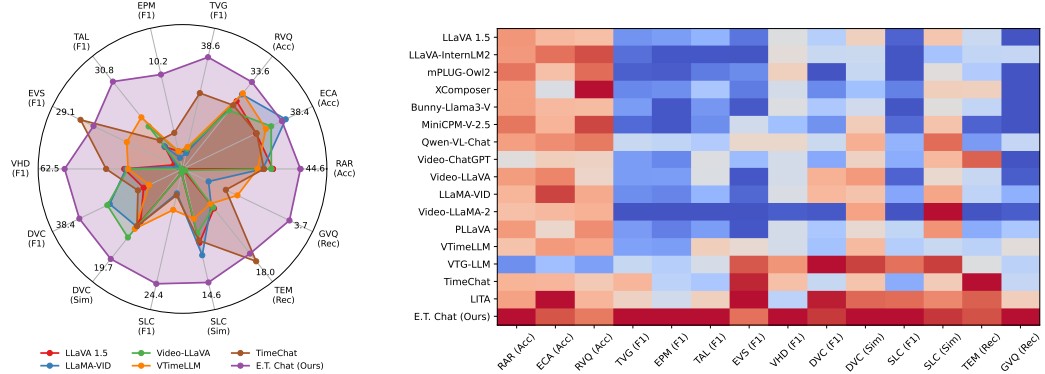

Figure 6: **Left:** Performance comparison between E.T. Chat and representative models. **Right:** Ranking of MLLMs on E.T. Bench, where red means higher ranks and blue represents lower ranks.

**E.T. Chat fills the gap between open-source and commercial MLLMs.** Benefit from the novel timestamps processing design and the multi-event instruction-tuning data, E.T. Chat achieves state-of-the-art performance among open-source MLLMs on most tasks, and obtain comparable results as commercial MLLMs. Notably, significant improvements can be viewed on [EPM], [VHD], [TEM], and [GVQ]. Implementation details and ablation studies can be viewed in Section B.2 and Section D.4, respectively.

## 5    Related Work

**Video Large Language Models.** Video-LLMs [66, 53, 51, 110, 86, 82, 34] represent a class of intelligent chatbots capable of understanding videos and perform various open-ended tasks. Generally, a Video-LLM comprises a visual encoder [77, 109] for perception, a projector [37, 46] for feature alignemnt, and a LLM [94, 94, 18, 11] for reasoning and response generation. VideoChat [47] and Video-ChatGPT [66] are two earliest attempts in this direction. Following works tend to provide better solutions via adding audio modality [110], joint training on images and videos [53, 40], or performing alignment before projection [53]. A recent trend [33, 82, 34, 76] involves the integration of Video-LLMs with time-sensitive understanding capabilities, while their solutions remain sub-optimal. Therefore, we propose to reframe timestamp generation as an embedding matching problem.

**Benchmarks for Video-LLMs.** The increasing number of Video-LLMs motivate the development of benchmarks [45, 16, 69, 67, 48, 63]. Among the earliest is SEED-Bench [45], a MLLM benchmark that supports both Image-LLMs and Video-LLMs and offers three evaluation dimensions in the realm of temporal modeling. AutoEval-Video [16] and Video-Bench [69] are designed specifically for videos. They employ LLMs for either QA generation or model evaluation. MVbench [48] provides a novel scheme to repurpose existing datasets for Video-LLM evaluation. Recent benchmarks also expand their scope and consider the ability of understanding extremely long videos [67, 86] or comprehending fine-grained temporal order information [63]. Nevertheless, none of the benchmarks have been designed for multi-event and time-sensitive understanding. In response to this gap, we introduce E.T. Bench, the first benchmark providing comprehensive evaluations on these scenarios.

## 6    Conclusion

In this work, we introduce **E.T. Bench**, a large-scale and comprehensive benchmark for multi-event & time-sensitive video-language understanding. Our benchmark encompasses a wide range of tasks on diverse video domains, evaluating multiple capabilities of Video-LLMs. Our experimental results reveal that current model designs and instruction-tuning data for Video-LLMs exhibit limitations in their capacity for timestamp representation and fine-grained multi-event modeling. To address these challenges, we further develop a novel model **E.T. Chat**, in conjunction with a multi-event instruction-tuning dataset, **E.T. Instruct 164K**, which serves as a robust baseline solution for such scenarios. We hope that the proposed benchmark, model, and instruction-tuning dataset will inspire future research on developing Video-LLMs.

## Acknowledgements

This work was supported in part by Hong Kong Research Grants Council GRF-15229423.

# Appendix

In the appendix, we provide more details about the proposed benchmark, model, and instruction-tuning dataset to complement the main paper. Additional analysis, ablation studies, visualizations, and discussions are also incorporated. Below is the table of content.

## A  Benchmark

### A.1  Pre-filtering Criteria

Table 4 presents the pre-filtering criteria for each source dataset when generating E.T. Bench.

### A.2  Annotation Repurposing and Cleaning

We summarize the detailed annotation repurposing and cleaning process for each task as follows.

[RAR] **Referred Action Recognition.** We adopt the *action localization* subset of Perception Test [74] for this task. We first select videos with at least three different actions, in which one action is sampled as ground truth. We then sample two other actions from the same video as intra-video distracters, and one action from other videos as inter-video distracter. The coarse timestamp hint is sampled from the segment containing only the ground truth action.

[ECA] **Event Caption Alignment.** We utilize Charades-STA [24] as data source. For each event-query pair, we randomly generate distracters (temporal boundaries) with $0.5\times$ to $2\times$ lengths compared with the ground truth, and ensure the temporal IoUs between any two options are no more than 0.5.

[RVQ] **Referred Video Question-Answering.** We leverage the high quality QA pairs from *interaction* and *sequence* question types of STAR [102]. Since the original annotations only contain question-relevant temporal boundaries, we randomly pick 20% of the QA pairs and modify their boundaries to have no overlap with the original ones, in order to synthesis the case when the question cannot be answered within the given boundary. An extra "unable to answer" option is added to all QA pairs.

Table 4: **Pre-filtering criteria for E.T. Bench generation.** *N/A* means no filtering.

| Type | Task | Source | Domain(s) | Selection Criteria |
|---|---|---|---|---|
| Referring | [RAR] Referred Action Recognition | Perception Test [74] | · Tabletop 
 · Indoor Activities 
 · Egocentric | · *Video ∉ [TAL] ∪ [TEM]* 
 · *Subset = "Action Localization"* 
 · *Video Duration ∈ [20s, 600s]* 
 · *#Actions per Video ⩾ 3* 
 · *Action Class ≠ "other"* |
| | [ECA] Event-Caption Alignment | Charades-STA [24] | · Indoor Activities | · *Video ∉ [TVG]* 
 · *Event Duration ∈ [2s, 30s]* |
| | [RVQ] Referred Video Question-Answering | STAR [102] | · Indoor Activities | · *QType ∈ {Interaction, Sequence}* |
| Grounding | [TVG] Temporal Video Grounding | QVHighlights [44] | · News & Vlogs | · *Video ∉ [VHD] ∪ [TEM]* 
 · *Moment Duration ∈ [2s, 50s]* 
 · *#Segments per Query = 1* |
| | | Charades-STA [24] | · Indoor Activities | · *Video ∉ [ECA]* 
 · *Moment Duration ∈ [2s, 50s]* |
| | [EPM] Episodic Memory | Ego4D-NLQ [26] | · Egocentric | · *Video Duration ⩽ 600s* 
 · *Moment Duration ∈ [3s, 50s]* |
| | [TAL] Temporal Action Localization | Perception Test [74] | · Tabletop 
 · Indoor Activities 
 · Egocentric | · *Video ∉ [RAR] ∪ [TEM]* 
 · *Subset = "Action Localization"* 
 · *Video Duration ∈ [20s, 600s]* 
 · *#Segments per Action ⩽ 10* 
 · *Action Class ∉ {moving object(s) around, other}* |
| | | THUMOS'14 [39] | · Sports | · *Video Duration ⩽ 600s* 
 · *#Segments per Action ⩽ 10* 
 · *Action Class ≠ "ambiguous"* |
| | | THUMOS'15 [25] | · Sports | · *Video Duration ⩽ 600s* 
 · *#Segments per Action ⩽ 10* 
 · *Action Class ≠ "ambiguous"* |
| | [EVS] Extractive Video Summarization | TVSum [87] | · Open | · *Summary Ratio ∈ [0.1, 0.25]* |
| | | SumMe [29] | · Open | · *Summary Ratio ∈ [0.1, 0.25]* |
| | [VHD] Video Highlight Detection | QVHighlights [44] | · News & Vlogs | · *Video ∉ [TVG] ∪ [TEM]* 
 · *#Highlights per Query ⩽ 2* |
| | | YouTube Highlights [89] | · Open | · *Highlight Ratio ∈ [0.05, 0.9]* |
| Dense Captioning | [DVC] Dense Video Captioning | HiREST [108] | · How To 
 · Indoor Activities 
 · Cooking | · *Query is Relevant to the Video* 
 · *Video is Clippable* |
| | | YouCook2 [113] | · Cooking | · *N/A* |
| | [SLC] Step Localization and Captioning | CrossTask [116] | · How To 
 · Cooking | · *Step Duration ⩾ 2s* 
 · *Repeated / Unordered Steps ∈ ∅* |
| | | HT-Step [3] | · How To 
 · Indoor Activities 
 · Cooking | · *Video Duration ∈ [10s, 600s]* 
 · *Step Duration ⩾ 2s* 
 · *#Steps per Video ⩾ 2* 
 · *Repeated / Unordered Steps ∈ ∅* |
| Complex Understanding | [TEM] Temporal Event Matching | Perception Test [74] | · Tabletop 
 · Indoor Activities 
 · Egocentric | · *Video ∉ [RAR] ∪ [TAL]* 
 · *Subset = "Action Localization"* 
 · *Video Duration ∈ [20s, 600s]* 
 · *Action Duration ∈ [2s, 50s]* 
 · *#Actions per Video ⩾ 2* 
 · *Action Class ≠ "other"* |
| | | QVHighlights [44] | · News & Vlogs | · *Video ∉ [TVG] ∪ [VHD]* 
 · *Segment Duration ∈ [2s, 50s]* 
 · *#Segments per Query ⩾ 2* |
| | [GVQ] Grounded Video Question-Answering | QAEgo4D [9] | · Egocentric | · *Segment Duration ∈ [2s, 50s]* |

[`TVG`] **Temporal Video Grounding.** Two datasets, *i.e.*, Charades-STA [24] and QVHighlights [44] are chosen. We filter out the samples with event duration shorter than 2s or longer than 50s, as these are generally noisy samples. On QVHighlights, only the samples with single events are chosen to align with our formulation.

[`EPM`] **Episodic Memory.** We employ Ego4D-NLQ [26] and conduct a thorough data cleaning & verification process. First, we perform a rule-based fix for noisy questions. Some common cases are: 1) Question starts with an additional "Query Text:" string. 2) Typos such as "I" → "i" or "l". 3) Unclear references such as "person x". We also found that some questions are ambiguous in the context of long videos, so we randomly crop the all the videos to 300-second long.

[`TAL`] **Temporal Action Localization.** We adopt Perception Test [74], THUMOS'14 [39] (`test` split), and THUMOS'15 [25] (`val` split). Videos with ambiguous action classes, *e.g.*, *moving object(s) around* and *other* in Perception Test, and *ambiguous* in THUMOS, are discarded. To reduce the difficulty, samples with more than 10 ground truth moments are filtered out as well.

[`EVS`] **Extractive Video Summarization.** We repurpose TVSum [87] and SumMe [29] to generate samples. In conventional video summarization, each frame is annotated with a probability of being the summary, which is incompatible with Video-LLMs that cannot strictly produce temporally-aligned frame-level scores, as can be seen in the near-random results in [82, 76]. Therefore, we reformulate it to predicting a set of temporal boundaries that compose the summary. Ground truths are obtained by sorting the frame-level scores and generate boundaries for consecutive frames with top-15% scores. When a video is annotated by multiple annotators (*i.e.*, in TVSum), we simply average the scores. This reformulation helps models persistently generate reasonable results.

[`VHD`] **Video Highlight Detection.** Samples are generated from QVHighlights [44] and YouTube Highlights [89] using a similar method as [`EVS`], but for highlights we only consider the frames with the highest scores. That means, ground truths are the frames with the highest highlight saliencies. During inference, a prediction is considered correct if the timestamp falls into any of the temporal boundaries. We directly utilize the text queries in QVHighlights as highlight queries. For YouTube Highlights, the domains are used, and videos with highlights covering more than 90% are discarded.

[`DVC`] **Dense Video Captioning.** We utilize YouCook2 [113] and HiREST [108]. Although HiREST only contains instructional videos, it is still considered as [`DVC`] rather than [`SLC`] because of the large event coverage. We also trimmed out the opening and closing scenes of HiREST videos.

[`SLC`] **Step Localization and Captioning.** We select CrossTask [116] and HT-Step [3] for this task. For CrossTask, we filter out the samples with wrong annotations, *e.g.*, repeated steps, and select only the videos with all steps longer than 2s to avoid ambiguous annotations. For HT-Step, we keep only the videos with at least 2 steps and remove the samples with incorrect temporal orders.

[`TEM`] **Temporal Event Matching.** We repurpose Perception Test [74] and QVHighlights [44] for this novel task, where the former focus on actions and the latter is for general events. We select videos with actions (excluding the *other* category) occurs multiple times from Perception Test, and sample one temporal boundary as the input reference. Other boundaries are used as ground truths. For QVHighlights, samples with one query referring to multiple disjoint moments are used.

[`GVQ`] **Grounded Video Question-Answering.** We adopt QAEgo4D [9] which naturally contains both QA pairs and corresponding timestamps derived from Ego4D-NLQ [26]. To control the task difficulty, we randomly crop all the videos to 150-second long. Typos in QA pairs are fixed.

### A.3 Instruction Templates

The instruction templates for different tasks are shown in Table 5. To ensure the models give responses in desired formats, each instruction starts with a sentence introducing the domain/title of the video, followed by detailed requirements about the task. We also add an explicit statement about the format of response and an example as guidance. The model outputs are passed through carefully designed rule-based parsers for answer extraction before evaluation.

### A.4 Distribution of Queries

We visualize the frequency distribution of verbs and nouns in E.T. Bench in Figure 7 and Figure 8, respectively. The distribution histograms demonstrate the diversity of queries in E.T. Bench.

Table 5: **Instruction templates in E.T. Bench.** Green text indicates the candidate expressions for difference domains. Blue text means the placeholder to be filled according to each sample. <time> denotes the timestamp representation, *e.g.*, *"23.6s"* (for text only models) or special time tokens.

| Type | Task | Instruction Template | Example Response |
|------|------|---------------------|------------------|
| Referring | [RAR] | You are given a video containing a series of actions. Watch the video carefully and identify the action around <time> by choosing from a set of options. The format of your response should be: "Best Option: (your choice)". For example: "Best Option: (B)". Now I give you the options: (A) <option> (B) <option> (C) <option> (D) <option>. Please provide your choice. | Best Option: (A) |
| | [ECA] | You are given a video about indoor activities. Watch the video carefully and select the moment that can be best described by the sentence "<query>". The format of your response should be: "Best Option: (your choice)". For example: "Best Option: (A)". Now I give you the options: (A) <time> - <time> (B) <time> - <time> (C) <time> - <time> (D) <time> - <time>. Please provide your choice. | Best Option: (C) |
| | [RVQ] | You are given a video about indoor activities. Watch the video carefully and answer a multiple choice question solely based on the event in <time> - <time>. The format of your response should be: "Best Option: (your choice)". For example: "Best Option: (C)". You may select "unable to answer" if the question can not be answered based on the provided moment. Now I give you the question: "<question>". The options are (A) <option> (B) <option> (C) <option> (D) <option> (E) <option>. Please provide your choice. | Best Option: (D) |
| Grounding | [TVG] | You are given a video about daily activities / indoor activities. Watch the video carefully and find a visual event described by the sentence: "<query>". The format of your response should be: "The event happens in <start time> - <end time>". You must represent start and end times in seconds. For example: "The event happens in 10.2 - 12.8 seconds". | The event happens in <time> - <time>. |
| | [EPM] | You are given an egocentric video about daily activities. Watch the video carefully and find a visual event that can answer the question: "<question>". The format of your response should be: "The event happens in <start time> - <end time>". You must represent start and end times in seconds. For example: "The event happens in 10.2 - 12.8 seconds". | The event happens in <time> - <time>. |
| | [TAL] | You are given a video containing a series of actions. Watch the video carefully and find all the visual events belonging to the action category: "<action>". The format of your response should be: "The action happens in <start time> - <end time>, <start time> - <end time>, and <start time> - <end time>". You must represent start and end times in seconds. For example: "The action happens in 4.2 - 6.8, 7.5 - 10.3, 15.1 - 18.6, and 23.4 - 27.5 seconds". | The action happens in <time> - <time> and <time> - <time>. |
| | [EVS] | You are given a video about <domain>. Watch the video carefully and summarize it into multiple short segments. The total length of the segments should be about 15% of the original video. The format of your response should be: "The summary locates in <start time> - <end time>, <start time> - <end time>, and <start time> - <end time>". You must represent start and end times in seconds. For example: "The summary locates in 5.2 - 7.5, 9.4 - 12.3, 16.9 - 18.2, and 21.8 - 25.4 seconds". | The summary locates in <time> - <time> and <time> - <time>. |
| | [VHD] | You are given a video about daily activities. Watch the video carefully and find a highlight moment according to the sentence / its domain: "<query>". The format of your response should be: "The highlight moment happens at <time>". You must represent time in seconds. For example: "The highlight moment happens at 26.8 seconds". | The highlight moment happens at <time>. |
| Dense Captioning | [DVC] | You are given a video about "<query>". Watch the video carefully and densely describe all the events in it. For each event, you need to determine the start and ends times and provide a concise description. The format of your response should be: "<start time> - <end time>, <description>". For example: "90 - 102 seconds, spread margarine on two slices of white bread. 114 - 127 seconds, place a slice of cheese on the bread.". | <time> - <time>, place bulgur wheat in a bowl and add boiling water. <time> - <time>, finely chop a bundle of parsley and add to a bowl. <time> - <time>, remove the leaves from stalks of mint chop finely and add to the parsley. |
| | [SLC] | You are given a video about "<task>". Watch the video carefully and identify all the key steps in it. For each step, you need to determine the start and ends times and provide a concise description using a few words. The format of your response should be: "<start time> - <end time>, <description>". You must represent start and end times in seconds. For example: "24.8 - 30.2 seconds, cut apple. 35.6 - 40.4 seconds, wash dishes.". | <time> - <time>, add sugar. <time> - <time>, pour water. <time> - <time>, cut lemon. <time> - <time>, squeeze lemon. <time> - <time>, pour lemon juice. <time> - <time>, stir mixture. |
| Complex Understanding | [TEM] | You are given a video about daily activities / containing a series of actions. Watch the video carefully and identify the event in <time> - <time>, then localize a different moment that contains the most similar event. The format of your response should be: "The similar event happens in <start time> - <end time>". You must represent start and end times in seconds. For example: "The similar event happens in 16.8 - 20.4 seconds". | The similar event happens in <time> - <time>. |
| | [GVQ] | You are given an egocentric video about daily activities. Watch the video carefully and answer a multiple choice question. Your answer should contain a choice of the best option and a relevant moment that supports your answer. The format of your response should be: "Best Option: (your choice). The relevant event happens in <start time> - <end time>". Now I give you the question: "<question>". The options are (A) <option> (B) <option> (C) <option> (D) <option>. Please provide your choice and the relevant moment. | Best Option: (C). The relevant event happens in <time> - <time>. |

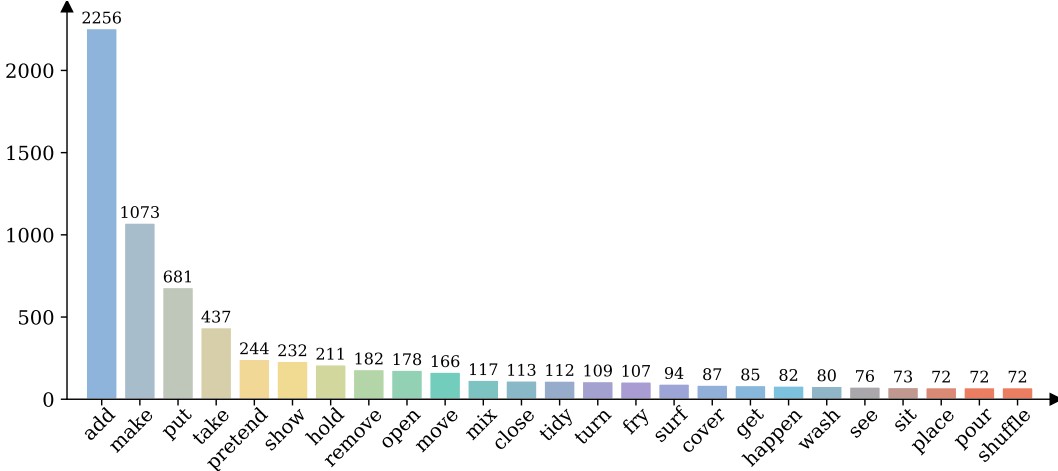

Figure 7: **Frequency distribution of verbs in E.T. Bench.** We only visualize the top 25 out of 461 verbs for clarity. The x- and y-axes denote the verbs and their frequencies, respectively.

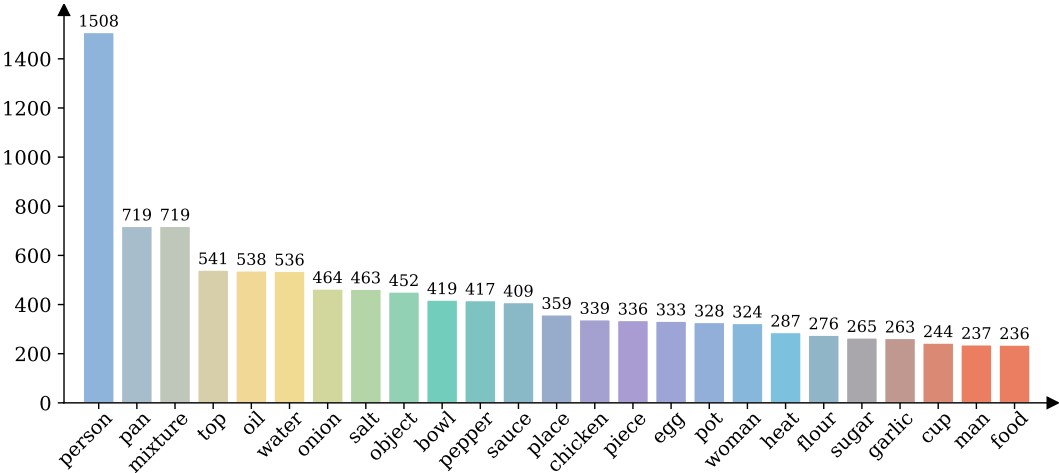

Figure 8: **Frequency distribution of nouns in E.T. Bench.** We only visualize the top 25 out of 3,090 nouns for clarity. The x- and y-axes denote the nouns and their frequencies, respectively.

# B Method

## B.1 Design Space for Timestamp Processing

As illustrated in Figure 9, existing MLLMs typically handle timestamps (for videos) or coordinates (for images) in three ways: 1) **Numerical Expressions** [75, 13, 82, 33]: Representing numbers directly in the form of text. This straightforward strategy loses continuity among numbers [21, 23]. A wrong prediction could be extremely far away from the ground truth. 2) **Special Tokens** [15, 99, 34, 76]: Defining a set of special tokens to quantize time/position into a fixed number (typically 100 to 300) of bins. This solution inevitably brings severe quantization loss, and it is not flexible for videos with variable lengths. Moreover, introducing too many new tokens into the vocabulary would break the pre-trained distribution of LLMs, making them hard to optimize without post pre-training. 3) **External Modules** [43, 65, 30]: Leveraging pre-trained external models (*e.g.*, SAM [41]) for grounding. This would introduce extra parameters and latency to LLMs. It is not directly compatible with videos as well, as existing temporal grounding models [44, 61, 68, 57, 60] are domain-specific and hard to generalize to all scenarios like SAM. Therefore, we propose to reformulate timestamp prediction as an *embedding matching* problem.

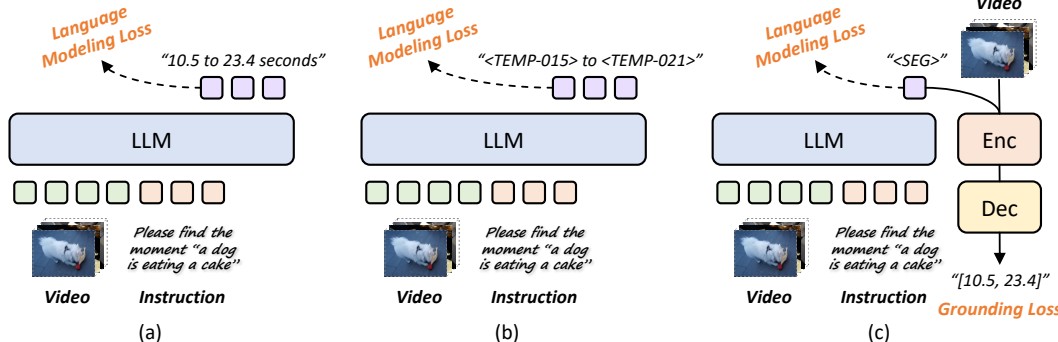

Figure 9: **Design space for timestamp processing.** Existing MLLMs handle timestamps in videos or coordinates in images via a) numerical expressions, b) special tokens, or c) external modules. Details are discussed in Section B.1.

## B.2 Implementation Details

We adopt the pre-trained ViT-G/14 from EVA-CLIP [22] as visual encoder. The architecture of frame compressor and LLM are based on Q-Former [19, 46] and Phi-3-Mini-3.8B [1], respectively. We first pre-train the model following the stage-1 and stage-2 recipes in [51], then activate the MLP projectors $E_{vid}$ & $E_{frm}$ and fine-tune them on E.T. Instruct 164K. $E_{vid}$ and $E_{frm}$ are randomly initialized, and the embeddings for <vid> token are initialized from the averaged embeddings of all existing tokens. During fine-tuning, we freeze the visual encoder and the FFN layers in Q-Former, and introduce LoRA [32] adapters on the LLM. Therefore, only the attention layers and projectors in frame compressor $E_c$, LoRA adapters, and matching projectors ($E_{vid}$ & $E_{frm}$) are learnable. We train the model with mixed precision (FP16) on a compute node with $8 \times$ NVIDIA V100 GPUs. The training process costs around 20 hours. More detailed hyper-parameters are listed in Table 6.

## C Instruction-Tuning Dataset

To fill the gap of lacking multi-event and time-sensitive training data for MLLMs, we introduce E.T. Instruct 164K, a large-scale instruction-tuning dataset with fine-grained timestamp annotations. Statistics about the dataset are shown in Table 7.

### C.1 Task Selection

Table 6: **Hyper-parameters for fine-tuning.**

| Hyper-parameter | Value |
|---|---|
| *Visual Encoder $E_v$* | |
| Frame Sampling Rate | 1 FPS |
| Preprocessing | Center Crop |
| Input Resolution | $224 \times 224$ |
| Patch Size | $14 \times 14$ |
| *Frame Compressor $E_c$* | |
| Number of Learnable Queries $M$ | 32 |
| Number of Layers | 12 |
| Hidden Size | 768 |
| *MLP Projector $E_{vid}$ & $E_{frm}$* | |
| Number of Layers | 2 |
| Hidden Size | 1536 |
| Output Size | 3072 |
| *Large Language Model* | |
| LoRA $r$ | 128 |
| LoRA $\alpha$ | 256 |
| LoRA Dropout Rate | 0.05 |
| LoRA Modules | QKVO Layers |
| *Model Training* | |
| Max Number of Tokens | 2048 |
| Number of Epochs | 1 |
| Batch Size | 32 |
| Learning Rate for LoRA | 5e-5 |
| Learning Rate for Other Parameters | 2e-5 |
| Weight Decay | 0.0 |
| Warmup Ratio | 0.03 |
| LR Scheduler Type | Cosine |
| Optimizer | AdamW [64] |
| AdamW $\beta_1$, $\beta_2$ | (0.9, 0.999) |

E.T. Instruct 164K covers 9 event-level understanding tasks, including [RVC] referred video captioning, [TVG] temporal video grounding, [TAL] temporal action localization, [EVS] extractive video summarization, [VHD] video highlight detection, [DVC] dense video captioning, [TVC] tagged video captioning, [SLC] step localization and captioning, and [GVQ] grounded video question-answering. Most tasks are aligned with E.T. Bench but with different source datasets. The only exception are [RVC] and [TVC], where the former requires the model to generate captions for the the given temporal boundary, and the latter is similar to [DVC] but with starting timestamps only. Note that [RAR], [ECA], [RVQ], [EPM], and [TEM] are in E.T. Bench only, which can be regarded as held-out tasks during evaluation.

Table 7: **Task and sample distribution in E.T. Instruct 164K.**

| Task | Source | Manual Label | Avg. Duration | #Samples | Ratio |
|------|--------|:---:|:---:|---:|---:|
| [RVC] | HowToCaption [84] | ✗ | 176.9s | 16,907 | 10.3% |
| [TVG] | DiDeMo [5] | ✓ | 49.0s | 33,000 | 20.1% |
| | QueryD [70] | ✗ | 173.7s | 4,267 | 2.6% |
| | TACoS [79] | ✓ | 151.9s | 6,693 | 4.1% |
| | NaQ [78] | ✗ | 296.5s | 10,546 | 6.4% |
| [TAL] | ActivityNet [10] | ✓ | 118.8s | 9,807 | 6.0% |
| | HACS [112] | ✓ | 159.5s | 15,218 | 9.3% |
| [EVS] | VideoXum [55] | ✓ | 123.5s | 7,989 | 4.9% |
| [VHD] | Mr. HiSum [88] | ✗ | 196.4s | 9,056 | 5.5% |
| [DVC] | ActivityNet Captions [42] | ✓ | 118.9s | 9,830 | 6.0% |
| [TVC] | ViTT [35] | ✗ | 210.0s | 2908 | 1.8% |
| [SLC] | COIN [90] | ✓ | 138.9s | 7,659 | 4.7% |
| | HowToStep [52] | ✗ | 189.8s | 20,000 | 12.2% |
| [GVQ] | EgoTimeQA [20] | ✗ | 150.0s | 10,000 | 6.1% |
| Total | | | 146.4s | 163,880 | 100% |

## C.2 Data Collection and Instruction Generation

We meticulously sample videos and annotations from 14 datasets, including HowToCaption [84], DiDeMo [5], QueryD [70], TACoS [79], NaQ [78], ActivityNet [10], HACS [112], VideoXum [55], Mr. HiSum [88], ActivityNet Captions [42], ViTT [35], COIN [90], HowToStep [52], and EgoTimeQA [20]. During sampling, we ensure that the videos have no overlap with E.T. Bench. Different from E.T. Bench in which all the samples are manually labeled, more than 40% of the samples in E.T. Instruct 164K are with automatically generated annotations. A similar filtering and rule-based cleaning process as E.T. Bench generation is conducted. Note that videos from EgoTimeQA are randomly cropped to 150-second long to reduce ambiguity during training.

We then convert the original annotations into instruction-following formats. Following previous works [49, 82], for each task, we carefully write a well-designed instruction, then prompt GPT-4 [2] to extend it to multiple diverse expressions. For some tasks having overlap with previous work [82], existing instructions are also taken into consideration. We manually select and refine 6 expressions to serve as the instruction templates for each task. To obtain ground truth responses, we convert the original annotations into natural language styles using manually designed templates. The generated instruction templates and response formats are shown in Table 8 & 9.

## D Experiments

### D.1 Evaluation Metrics

**Referring.** All the tasks are formulated as MCQs, thus we adopt accuracy as the main metric.

**Grounding.** We compute F1 scores averaged among IoU thresholds $\theta_{IoU}$ at four levels (0.1, 0.3, 0.5, and 0.7). For [TVG] and [EPM], only the first predicted temporal boundary is accepted. This aligns with conventional settings that use Recall@1 as metrics. For [TAL], all the predicted boundaries are used. In [EVS], F1 scores are computed at clip level, that is, each video is divided into 1-second long clips, and precision/recall is defined as the percentage of true positive clips with respect to all the predicted/ground truth clips. For [VHD], a prediction (single timestamp) is regarded as a true positive when it falls within any of the ground truth boundaries.

**Dense Captioning.** Similar to *grounding*, we utilize F1 score at the same four levels of $\theta_{IoU}$ for boundary predictions in [DVC] and [SLC]. This also aligns with previous works in these areas [42, 98]. To measure the correctness of descriptions, previous works leverage traditional metrics [73, 54, 8, 97] for machine translation, which cannot handle ambiguity in open-ended scenarios. Therefore, we instead perform evaluation at semantic level and employ sentence similarity [81] to

Table 8: **Instruction and response templates in E.T. Instruct 164K (Part I).** Blue text means the placeholder to be filled according to each sample, and <time> denotes the timestamp representation.

| Task | Instruction Templates | Response Format |
|---|---|---|
| [RVC] | · Watch the video segment in <time> - <time> and provide a concise description for it.
· Watch the video portion spanning from <time> to <time> and provide a depiction of its main event.
· Describe the video event happened in <time> - <time>.
· Depict the event happening in the video from <time> to <time>.
· Provide a description of the activity shown in the video within <time> - <time>.
· Focus on the video event in <time> - <time> and describe what is happening. | <answer>. |
| [TVG] | · Localize the visual content described by the given textual query "<query>" in the video, and output the start and end timestamps in seconds.
· Detect and report the start and end timestamps of the video segment that semantically matches the given textual query "<query>".
· Give you a textual query: "<query>". When does the described content occur in the video? Please return the timestamp in seconds.
· Locate the visual content mentioned in the text query "<query>" within the video using timestamps.
· The given natural language query "<query>" is semantically aligned with a video moment, please give the start time and end time of the video moment.
· Find the video segment that corresponds to the given textual query "<query>" and determine its start and end seconds. | The event happens in <time> - <time>. |
| [TAL] | · Detect and localize all the video segments containing the given action "<action>", and provide the outputs using start and end timestamps.
· Find all the sections of the video where the action "<action>" occurs, and report the results with their respective start and end timestamps.
· For the given action "<action>", locate all corresponding video sections and present the results with starting and ending timestamps.
· Discover and determine the locations of all video portions containing the action "<action>", and return the start and end time in seconds.
· Locate all instances of the action "<action>" in the video and give me the start and end times for each occurrence.
· Identify and list all segments of the video where the action "<action>" takes place, providing the start and end times for each instance. | The action happens in <time> - <time>, <time> - <time>, and <time> - <time>. |
| [EVS] | · Summarize the video to about 15% of the original length based on the video domain "<domain>", and provide the outputs using start and end timestamps.
· Condense the video to approximately 15% of its original length, focusing on the domain "<domain>", and include start and end timestamps for each summarized segment.
· Reduce the video's length to roughly 15%, emphasizing the "<domain>" domain, and provide the outputs with their respective start and end timestamps.
· Generate a 15% summary of the video, particularly related to the domain "<domain>", and include the start and end times for each selected portion.
· Provide a summary of the video, reducing it to around 15% of its length, with a specific focus on the domain "<domain>", including start and end timestamps for the selected segments.
· Summarize the video to about 15% of its total duration, specifically highlighting the domain "<domain>", and include start and end timestamps for each segment. | The summary locates in <time> - <time>, <time> - <time>, and <time> - <time>. |
| [VHD] | · Find a highlight moment in the video according to the given query: "<query>", and return its timestamp.
· Identify a highlight in the video that matches the query "<query>", and provide the corresponding timestamp.
· Detect a highlight in the video corresponding to the query "<query>", and report its timestamp.
· Identify a highlight moment in the video based on "<query>", and give me the timestamp.
· Localize a highlight moment that matches the given query "<query>", and return the timestamp for it.
· According to the query "<query>", please find a single highlight moment in the video and provide its timestamp. | The highlight moment happens at <time>. |
| [DVC] | · Localize a series of activity events in the video, output the start and end timestamp for each event, and describe each event with sentences. The output format of each predicted event should be like: "start - end seconds, event description".
· Determine the start and end times of various activity events in the video, accompanied by descriptions.
· Capture and describe the activity events in the given video, specifying their respective time intervals, and output the time intervals in the "start - end seconds format".
· Identify, timestamp, and describe various activity events occurring in the video. The timestamp should include the start time and end time in seconds.
· Detect and report the start and end timestamps of activity events in the video, along with descriptions.
· Pinpoint the time intervals of activity events in the video, and provide detailed descriptions for each event. | <time> - <time>, <event>. <time> - <time>, <event>. <time> - <time>, <event>. |

Table 9: **Instruction and response templates in E.T. Instruct 164K (Part II).** Blue text means the placeholder to be filled according to each sample, and <time> denotes the timestamp representation.

| Task | Instruction Templates | Response Format |
|------|----------------------|-----------------|
| [TVC] | · Densely capture all the events happened in the video, and describe them in the form of "start time, description".
· List all the events in the video in detail and format them as "start time, description".
· Chronologically describe each event in the video, noting "start time, description".
· Identify all the events in the video and provide a concise description for each of them. The response for each event should contain its start time and the description.
· Pinpoint all events in the video and give a concise description with their respective start times.
· Recognize every event in the video, describing each concisely and report it as "start time, description". | <time>, <event>. <time>, <event>. <time>, <event>. |
| [SLC] | · Localize a series of action steps in the given video, output a start and end timestamp for each step, and briefly describe the step.
· Locate and describe a series of actions or steps in the video, including their start and end timestamps.
· Identify and mark the video segments corresponding to a series of actions or steps, specifying the timestamps and describing the steps.
· Find, identify, and determine the temporal boundaries of a series of distinct actions or steps occurring throughout the video. For each action, output the corresponding start and end timestamps, accompanied by a concise description.
· Identify and localize a series of steps or actions occurring in the video, providing start and end timestamps and related descriptions.
· Locate and pinpoint a sequential series of specific actions or steps in the video, accurately specifying the start and end timestamps for each action. Additionally, provide a succinct description. | <time> - <time>, <step>. <time> - <time>, <step>. <time> - <time>, <step>. |
| [GVQ] | · Watch the video carefully and answer the question: "<question>". Your response should mention the start and end timestamps as a reference. For example: "<answer>. The relevant event happens in <start time> to <end time>".
· Given the question: "<question>", please watch the video carefully and provide both the answer and the relative moment that as a reference.
· Taking the question: "<question>" into consideration, please watch the video attentively and provide your answer along with the exact timing as a reference.
· Answer the following question and provide the corresponding start and end timestamps depicting the relevant moment: "<question>".
· Give a response to the question "<question>" according to the video and include the precise times marking the relevant moment.
· After watching the video, provide an answer to the following question "<question>" and point out the relevant start and end times in the video. | <answer>. The relevant event happens in <time> - <time>. |

measure the distances between model outputs and ground truths. Following previous practices [20], the `all-MiniLM-L6-v2` model in Sentence Transformers* library is used as the embedding model.

**Complex Understanding.** We adopt Recall@1 as the metric for both [TEM] and [GVQ]. The IoU thresholds are aligned with *grounding* and *dense captioning*. For [TEM], only the first predicted temporal boundary is accepted, and it is regarded as a true positive when it has the maximum IoU among all ground truths larger than the threshold. For [GVQ], aside from boundary prediction, the MCQ answer should also be correct for a successful recall.

With the unified evaluation metrics, we are able to average them and measure a model's general performance under each capability. To achieve this, we further define 5 averaged metrics: 1) $\text{Acc}_{ref}$: Averaged accuracy on *referring* tasks; 2) $\text{F1}_{gnd}$: Averaged F1 score on *grounding* tasks; 3) $\text{F1}_{cap}$: Averaged F1 score on *dense captioning* tasks; 4) $\text{Sim}_{cap}$: Averaged sentence similarity on *dense captioning* tasks; 5) $\text{Rec}_{com}$: Averaged recall on *complex understanding* tasks. These metrics serve as indicators for general performance on event-level and time-sensitive video understanding.

## D.2 Baselines

We extensively evaluate 20 representative MLLMs on E.T. Bench, including 7 open-source Image-LLMs (LLaVA-1.5 [58], LLaVA-InternLM2 [11], mPLUG-Owl2 [105], InternLM-XComposer [111], Bunny-Llama3-V [31], MiniCPM-Llama3-V-2.5 [93], and Qwen-VL-Chat [7]), 9 open-source Video-LLMs (Video-ChatGPT [66], Video-LLaVA [53], LLaMA-VID [51], Video-LLaMA-2 [110], PLLaVA [104], VTimeLLM [33], VTG-LLM [28], TimeChat [82], and LITA [34]), and 4 commercial MLLMs (GPT-4V [71], GPT-4o [72], Gemini-1.5-Flash [80], and Gemini-1.5-Pro [80]). Note that the video interface for GPT-4o is not publicly available, hence we treat it as an Image-LLM instead.

---

*`https://github.com/UKPLab/sentence-transformers`

Table 10: **Model architectures of MLLMs evaluated on E.T. Bench.** Size means the LLM size.

| Model | Size | Frame Resolution | Sampled Frames | Visual Encoder | LLM |
|---|---|---|---|---|---|
| *Image-LLMs* | | | | | |
| LLaVA-1.5 [58] | 7B | $336 \times 336$ | 8 | CLIP-ViT-L/14 | Vicuna-1.5 |
| LLaVA-InternLM2 [11] | 7B | $336 \times 336$ | 8 | CLIP-ViT-L/14 | InternLM2 |
| mPLUG-Owl2 [105] | 7B | $448 \times 448$ | 8 | CLIP-ViT-L/14 | Llama-2 |
| InternLM-XComposer [111] | 7B | $224 \times 224$ | 8 | EVA-ViT-G/14 | InternLM |
| Bunny-Llama3-V [31] | 8B | $384 \times 384$ | 8 | SigLIP-ViT-L/14 | Llama-3 |
| MiniCPM-Llama3-V-2.5 [93] | 8B | $980 \times 980$ | 8 | SigLIP-ViT-L/14 | Llama-3 |
| Qwen-VL-Chat [7] | 7B | $448 \times 448$ | 8 | CLIP-ViT-bigG/14 | Qwen |
| *Video-LLMs* | | | | | |
| Video-ChatGPT [66] | 7B | $224 \times 224$ | 100 | CLIP-ViT-L/14 | LLaMA |
| Video-LLaVA [53] | 7B | $224 \times 224$ | 8 | LanguageBind-ViT-L/14 | Vicuna-1.5 |
| LLaMA-VID [51] | 7B | $224 \times 224$ | 1 FPS | EVA-ViT-G/14 | Vicuna-1.5 |
| Video-LLaMA-2 [110] | 7B | $224 \times 224$ | 8 | EVA-ViT-G/14 | Llama-2-Chat |
| PLLaVA [104] | 7B | $672 \times 672$ | 16 | CLIP-ViT-L/14 | Vicuna-1.5 |
| VTimeLLM [33] | 7B | $224 \times 224$ | 100 | CLIP-ViT-L/14 | Vicuna-1.5 |
| VTG-LLM [28] | 7B | $224 \times 224$ | 96 | EVA-ViT-G/14 | Llama-2 |
| TimeChat [82] | 7B | $224 \times 224$ | 96 | EVA-ViT-G/14 | Llama-2 |
| LITA [34] | 13B | $224 \times 224$ | 100 | CLIP-ViT-L/14 | Vicuna-1.3 |
| E.T. Chat (Ours) | 3.8B | $224 \times 224$ | 1 FPS | EVA-ViT-G/14 | Phi-3-Mini |

Table 11: **Comparison on architectural designs.**

| <vid> Token | Bi-directional | Smoothing | $\text{Acc}_{ref}$ | $\text{F1}_{gnd}$ | $\text{F1}_{cap}$ | $\text{Sim}_{cap}$ | $\text{Rec}_{com}$ |
|---|---|---|---|---|---|---|---|
| | | | 25.0 | 17.5 | 21.2 | 12.3 | 8.6 |
| ✓ | | | 34.0 | 25.2 | 26.4 | 15.4 | 9.2 |
| ✓ | ✓ | | 33.7 | 30.5 | 27.5 | 15.8 | 9.8 |
| ✓ | | ✓ | 34.5 | 25.8 | 26.4 | 13.6 | 9.5 |
| ✓ | ✓ | ✓ | **38.4** | **33.5** | **31.4** | **17.1** | **10.1** |

We compare the architectures of open-source MLLMs in Table 10. Optional visual encoders for these models are CLIP [77], EVA [22], SigLIP [109], OpenCLIP [36], and LanguageBind [114], while the LLM backbones include LLaMA [94], Llama-2 [95], Llama-3 [4], Vicuna [18], InternLM [92], InternLM2 [11], Qwen [6], and Phi-3-Mini [1].

## D.3 More Benchmark Results

In Table 19 and Table 20, we provide performance breakdown across source datasets, where we observe that the ranking of models differs across source datasets. Table $21 \sim 25$ present detailed comparisons under different IoU thresholds $\theta_{IoU}$ and more metrics (*e.g.*, METEOR [8], Rouge-L [54], and CIDEr [97]) on [TVG], [EPM], [TAL], [DVC], [SLC], [TEM], and [GVQ] tasks.

## D.4 Ablation Studies

**Effect of architectural designs.** We verify the effectiveness of <vid> token, bi-directional attention across video tokens, and label smoothing during training. The results are compared in Table 11. Without introducing the <vid> token (first row), our model falls back to the *numerical expression* variant discussed in Section B.1, which struggles in timestamp prediction even with instruction-tuning on E.T. Instruct 164K. By reformulating timestamp prediction as embedding matching (second row), our method significantly works better on all tasks on E.T. Bench. Extra modifications, *i.e.*, bi-directional attention (third row) and label smoothing (fourth row), further enhance the model to achieve better localization abilities, demonstrated by the substantial increase in $\text{F1}_{gnd}$.

Table 12: **Choices of frame compressor.**

| Method | $\text{Acc}_{ref}$ | $\text{F1}_{gnd}$ | $\text{F1}_{cap}$ | $\text{Sim}_{cap}$ | $\text{Rec}_{com}$ |
|---|---|---|---|---|---|
| Pooling | 29.6 | 25.5 | 21.1 | 11.3 | 9.1 |
| Q-Former | **38.4** | **33.5** | **31.4** | **17.1** | **10.1** |

Table 13: **Choices of layers for matching.** Layer$_{vid}$ and Layer$_{frm}$ are the index of LLM layer utilized for matching, *e.g.*, "−1" means using the final-layer hidden states.

| Layer$_{vid}$ | Layer$_{frm}$ | Acc$_{ref}$ | F1$_{gnd}$ | F1$_{cap}$ | Sim$_{cap}$ | Rec$_{com}$ |
|---|---|---|---|---|---|---|
| −1 | −1 | 38.2 | 32.2 | 31.0 | **17.2** | 9.6 |
| −1 | −2 | 37.8 | 32.5 | 30.8 | 16.5 | 9.9 |
| −2 | −1 | **38.4** | 33.5 | **31.4** | 17.1 | **10.1** |
| −2 | −2 | 37.6 | **33.8** | 31.2 | 16.7 | 9.7 |

Table 14: **Comparison on learnable modules.** ATTN and FFN represent the attention and feed-forward layers in Q-Former, respectively.

| Q-Former $E_q$ | | Aggregator $E_a$ | Projector $E_p$ | LLM (LoRA) | Acc$_{ref}$ | F1$_{gnd}$ | F1$_{cap}$ | Sim$_{cap}$ | Rec$_{com}$ |
|---|---|---|---|---|---|---|---|---|---|
| **ATTN** | **FFN** | | | | | | | | |
| ❄ | ❄ | 🔥 | 🔥 | ❄ | 36.1 | 29.2 | 27.1 | 14.1 | 8.7 |
| ❄ | ❄ | 🔥 | 🔥 | 🔥 | 37.3 | 30.5 | 28.3 | 15.0 | 9.6 |
| ❄ | 🔥 | 🔥 | 🔥 | 🔥 | 37.9 | 31.8 | 29.6 | 15.7 | 9.5 |
| 🔥 | ❄ | 🔥 | 🔥 | 🔥 | **38.4** | **33.5** | 31.4 | 17.1 | **10.1** |
| 🔥 | 🔥 | 🔥 | 🔥 | 🔥 | 37.5 | 29.1 | **32.2** | **17.5** | 9.3 |

**Choice of frame compressor.** Table 12 compares the performance of two design choices for frame compressor $E_c$, *i.e.*, a naive spatial pooling among frame patches $\mathbf{P}_t$ and a query-guided compression based on Q-Former [46]. The results confirm that employing Q-Former for frame compression proves to be a superior alternative.

**Choice of LLM layer for matching.** As articulated in the main paper, during matching, we utilize the second-last layer's hidden states for the <vid> token, while the final-layer's hidden states for frame tokens. This strategy take into consideration the small feature range of final-layer hidden states for the <vid> token [30]. We further verify its effectiveness in Table 13. While the results are essentially similar, current setting exhibits a marginally superior overall performance.

**Learnable modules.** In Table 14, we justify the training strategy for instruction-tuning. Updating only the context aggregator $E_a$ and projector $E_p$ (first row) makes the training hard to converge with new tokens, and fine-tuning the LLM with LoRA (second row) brings better performance. We contend that the pre-trained Q-Former is suitable only for short & single event videos due to the constraints of pre-training data. Serving as the frame compressor, such limitation would hinder the model's performance. Line $3 \sim 5$ corroborate our hypothesis, as updating Q-Former on E.T. Instruct 164K brings notable performance improvements. Furthermore, we observe that fine-tuning the whole Q-Former makes the model slightly overfit to dense captioning tasks, and freezing its FFN layers could strike the balance between adapting to new data and retaining pre-trained capabilities.

Table 15: **Effect of $\alpha$ for label smoothing.**

| $\alpha$ | Acc$_{ref}$ | F1$_{gnd}$ | F1$_{cap}$ | Sim$_{cap}$ | Rec$_{com}$ |
|---|---|---|---|---|---|
| 1.0 | 37.2 | 30.8 | 26.3 | 14.8 | 9.6 |
| 1.5 | 37.9 | 31.6 | 28.6 | 15.4 | **10.3** |
| 2.0 | **38.4** | 33.5 | **31.4** | **17.1** | 10.1 |
| 2.5 | 38.1 | 33.0 | 29.8 | 16.8 | 9.2 |
| 3.0 | 37.5 | **33.7** | 28.4 | 16.0 | 9.8 |

**Effect of $\alpha$ for label smoothing.** We ablate the effect of different $\alpha$ values for label smoothing in Table 15. Smaller $\alpha$ values make the optimization goal of matching scores smoother. Generally, setting $\alpha$ to around 2.0 brings considerable results.

**Joint effect of model and instruction-tuning dataset.** We compare in Table 16 the joint effect of model design and instruction-tuning dataset collection. We choose two representative models (LLaMA-VID [51] and TimeChat [82]) as baselines and train them on E.T. Instruct 164K. Our E.T. Chat is also trained on TimeIT dataset [82] for in-depth comparison. The comparison results between line 1 & 2, 3 & 4, and 5 & 6 demonstrate the effectiveness of E.T. Instruct 164K. Results in line 2, 4, and 6 verify the significance of our model design.

**Effect of instruction-tuning tasks.** To study the effect of each task during instruction tuning, we provide detailed comparisons in Table 17. We observe that adding more tasks for instruction-tuning

Table 16: **Joint effect of model and instruction-tuning (IT) dataset.**

| Model | IT Dataset | $\text{Acc}_{ref}$ | $\text{F1}_{gnd}$ | $\text{F1}_{cap}$ | $\text{Sim}_{cap}$ | $\text{Rec}_{com}$ |
|---|---|---|---|---|---|---|
| LLaMA-VID [51] | 723K Corpus [51] | 32.5 | 9.2 | 16.2 | 11.9 | 4.0 |
| LLaMA-VID [51] | E.T. Instruct 164K (Ours) | 31.3 | 16.0 | 19.8 | 14.9 | 7.8 |
| TimeChat [82] | TimeIT [82] | 27.7 | 21.9 | 11.1 | 10.8 | 9.7 |
| TimeChat [82] | E.T. Instruct 164K (Ours) | 29.5 | 24.3 | 21.5 | 11.5 | **11.4** |
| E.T. Chat (Ours) | TimeIT [82] | 34.9 | 22.1 | 20.1 | 13.4 | 6.9 |
| E.T. Chat (Ours) | E.T. Instruct 164K (Ours) | **38.4** | **33.5** | **31.4** | **17.1** | 10.1 |

Table 17: **Ablation study on instruction-tuning tasks.**

| RVC | TVG | TAL | EVS | VHD | DVC | SLC | GVQ | $\text{Acc}_{ref}$ | $\text{F1}_{gnd}$ | $\text{F1}_{cap}$ | $\text{Sim}_{cap}$ | $\text{Rec}_{com}$ |
|---|---|---|---|---|---|---|---|---|---|---|---|---|
| ✓ | | | | | | | | 36.5 | 9.8 | 0.4 | 10.3 | 0.5 |
| ✓ | ✓ | | | | | | | 36.0 | 12.8 | 3.7 | 12.5 | 5.1 |
| ✓ | ✓ | ✓ | | | | | | 35.4 | 31.9 | 10.3 | 11.5 | 9.9 |
| ✓ | ✓ | ✓ | ✓ | | | | | 35.6 | 32.5 | 9.5 | 11.8 | 9.5 |
| ✓ | ✓ | ✓ | ✓ | ✓ | | | | 35.9 | 33.6 | 15.1 | 10.8 | 9.7 |
| ✓ | ✓ | ✓ | ✓ | ✓ | ✓ | | | 37.4 | 34.8 | 18.2 | 12.5 | 9.4 |
| ✓ | ✓ | ✓ | ✓ | ✓ | ✓ | ✓ | | 34.2 | **33.7** | 28.5 | 14.3 | 9.2 |
| ✓ | ✓ | ✓ | ✓ | ✓ | ✓ | ✓ | ✓ | **38.4** | 33.5 | **31.4** | **17.1** | **10.1** |

might slightly affect the performance on original tasks. This can be alleviated by carefully balancing the number of samples per task.

### D.5 Qualitative Results

Figure 10 ∼ 15 present task-specific qualitative comparisons among 5 representative open-source MLLMs, *i.e.*, LLaVA-1.5 [58], Video-ChatGPT [66], LLaMA-VID [51], TimeChat [82], and E.T. Chat. The correct model responses are marked green. We observe that the unsatisfactory performance of existing methods comes from 1) weak instruction-following abilities, 2) low temporal resolution, 3) lack of event-level and time-sensitive designs, and 4) lack of multi-event instruction-tuning data.

## E Limitations and Future Work

Currently, the proposed E.T. Bench is based on `val` or `test` split of existing datasets, whose training split might be included for MLLM training. This could potentially result in data leakage, thereby compromising the integrity of the zero-shot evaluation framework and leading to unfair comparisons. Therefore, our next step would be self-collecting new videos and provide manual annotations under each carefully designed task. More flexible input-output formats shall also be incorporated to complement the existing benchmark.

For E.T. Chat, even with advanced frame compression strategies, the low spatial resolution (1 token per frame) limits the model's ability to understand spatial details. Modern Image-LLMs are becoming to support extra-high-resolution image inputs, but this is not directly compatible to videos due to the large compute resource consumption. Our future work will focus on the balance between spatial and temporal resolution for Video-LLMs.

## F Licenses

The annotations of E.T. Bench are provided to the public under CC BY-NC-SA 4.0 license. A copy can be obtained at `https://creativecommons.org/licenses/by-nc-sa/4.0/`. By downloading our dataset from our website or other sources, the user agree to adhere to the terms of CC BY-NC-SA 4.0 and licenses of the source datasets. Licenses of the source datasets are listed in Table 18.

Table 18: **Licenses of source datasets in E.T. Bench.**

| Dataset | License | Link |
|---------|---------|------|
| Perception Test [74] | CC BY 4.0 | https://creativecommons.org/licenses/by/4.0/ |
| Charades-STA [24, 85] | Non-Commercial Use | https://prior.allenai.org/projects/data/charades/license.txt |
| STAR [102] | Apache License 2.0 | https://github.com/csbobby/STAR/blob/main/LICENSE |
| QVHighlights [44] | CC BY-NC-SA 4.0 | https://creativecommons.org/licenses/by-nc-sa/4.0/ |
| Ego4D-NLQ [26] | Custom | https://ego4d-data.org/pdfs/Ego4D-Licenses-Draft.pdf |
| THUMOS'14 [39] | Research Purpose Only | https://www.crcv.ucf.edu/THUMOS14 |
| THUMOS'15 [25] | Research Purpose Only | http://www.thumos.info/ |
| TVSum [87] | CC BY 4.0 | https://creativecommons.org/licenses/by/4.0/ |
| SumMe [29] | N/A | https://doi.org/10.1007/978-3-319-10584-0_33 |
| YouTube Highlights [89] | N/A | https://doi.org/10.1007/978-3-319-10590-1_51 |
| HiREST [108] | MIT License | https://opensource.org/license/mit |
| YouCook2 [113] | MIT License | https://opensource.org/license/mit |
| CrossTask [116] | BSD 3-Clause License | https://opensource.org/license/bsd-3-clause |
| HT-Step [3] | CC BY-NC 4.0 | https://creativecommons.org/licenses/by-nc/4.0/ |
| QAEgo4D [9] | N/A | https://doi.org/10.1109/CVPRW56347.2022.00162 |

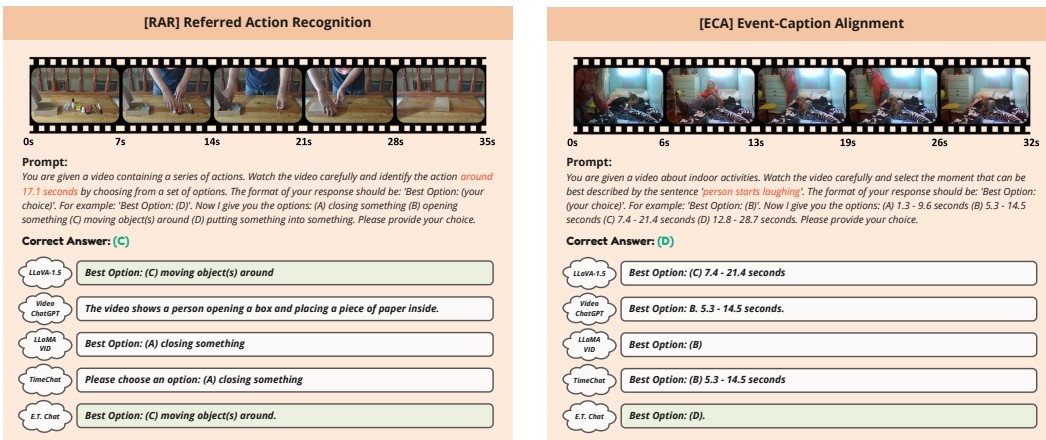

Figure 10: **Qualitative comparison on** `[RAR]` **(left) and** `[ECA]` **(right).**

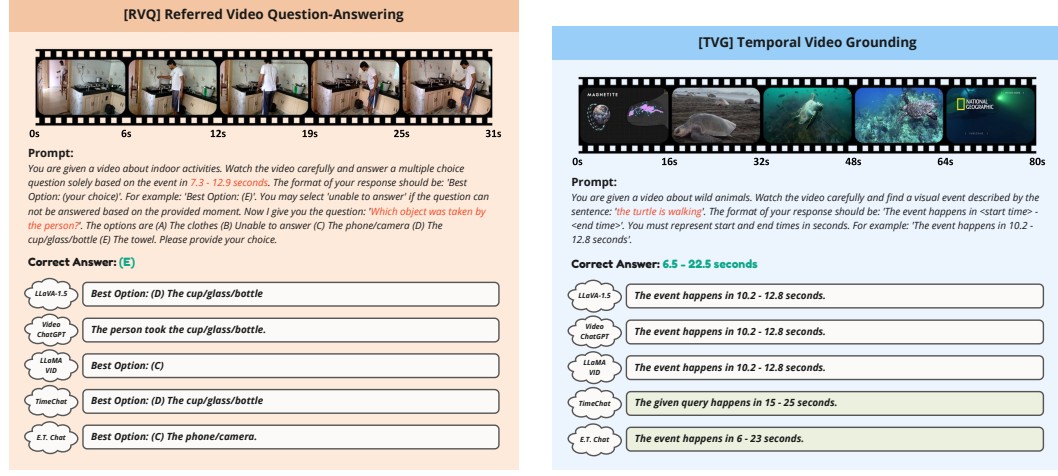

Figure 11: **Qualitative comparison on** `[RVQ]` **(left) and** `[TVG]` **(right).**

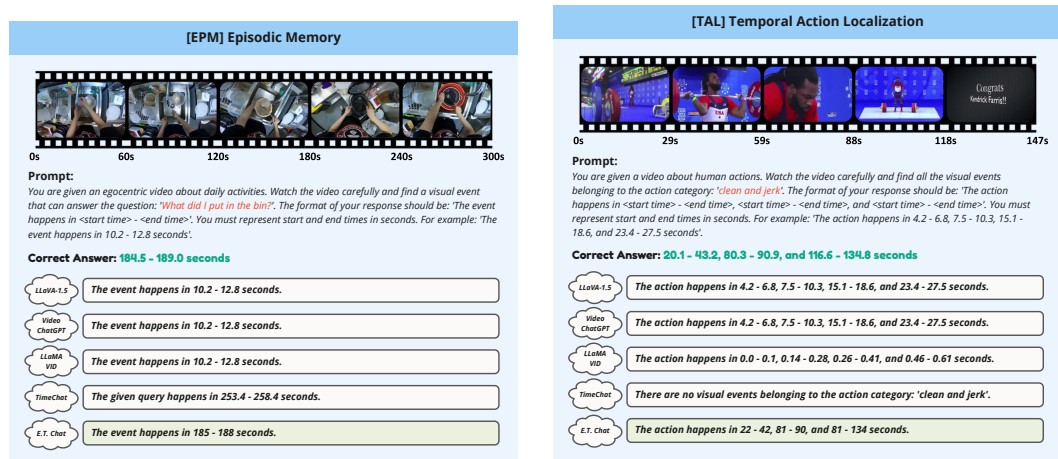

Figure 12: **Qualitative comparison on** [EPM] **(left) and** [TAL] **(right).**

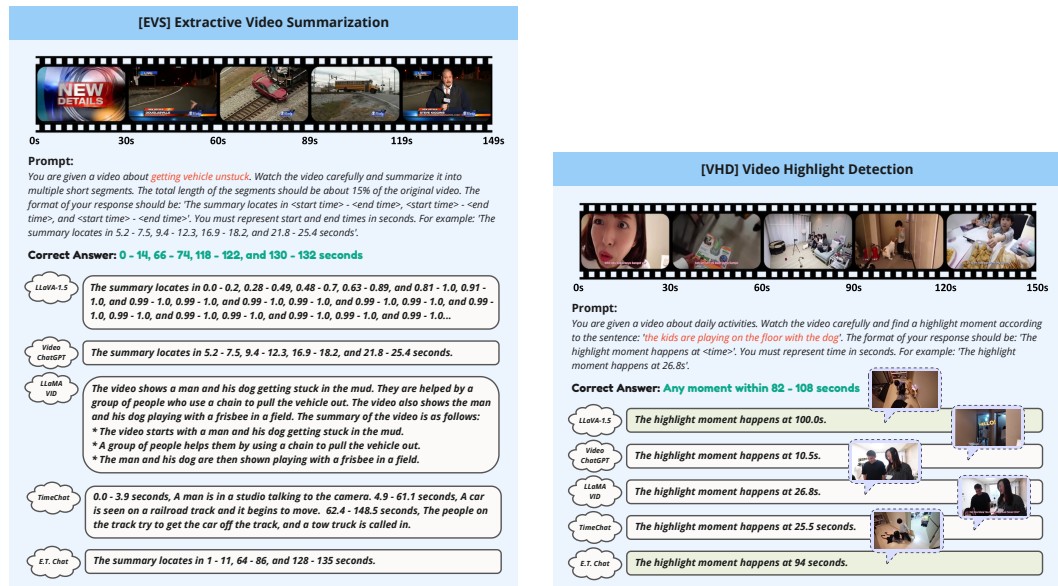

Figure 13: **Qualitative comparison on** [EVS] **(left) and** [VHD] **(right).**

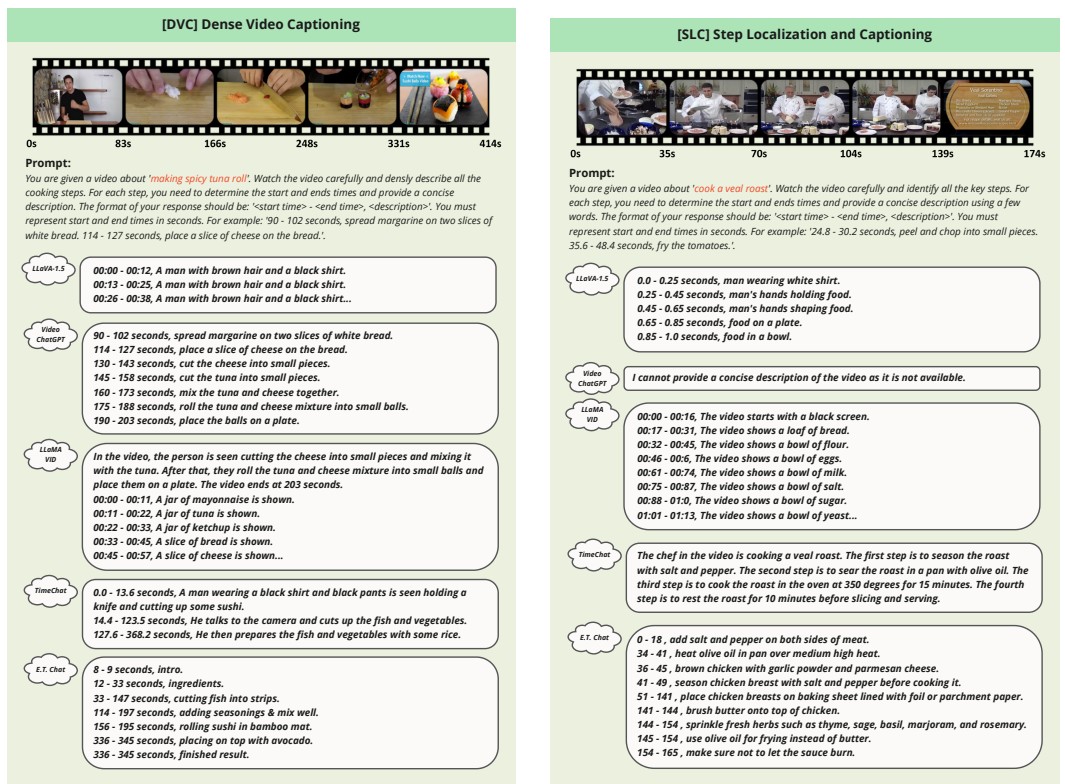

Figure 14: **Qualitative comparison on** [DVC] **(left) and** [SLC] **(right).**

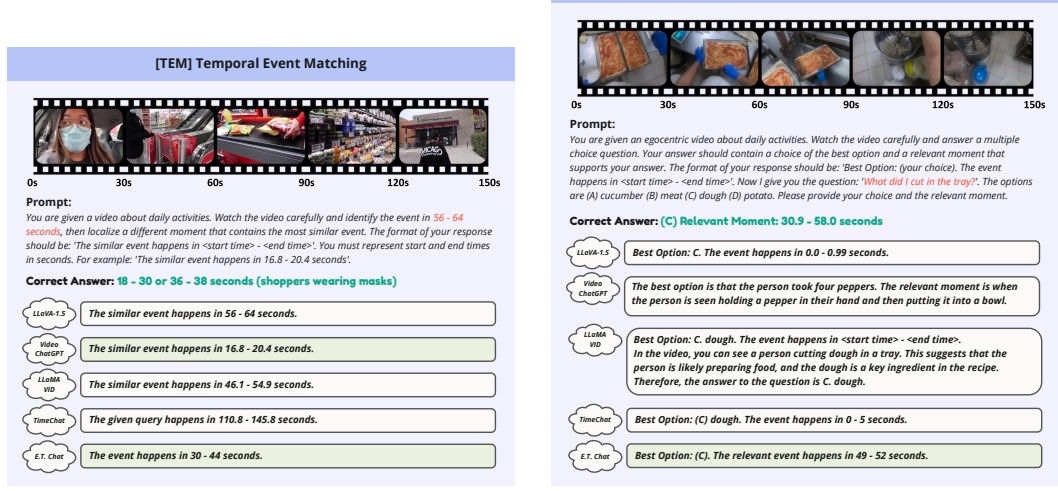

Figure 15: **Qualitative comparison on** [TEM] **(left) and** [GVQ] **(right).**

Table 19: **Performance breakdown across source datasets on *grounding* tasks.** Abbreviations: [QV] QVHighlights, [CS] Charades-STA, [EN] Ego4D-NLQ, [PT] Perception Test, [T14] THU-MOS'14, [T15] THUMOS'15, [SM] SumMe, [TV] TVSum, [YH] YouTube Highlights.

| Method | TVG | | EPM | TAL | | | EVS | | VHD | |
|---|---|---|---|---|---|---|---|---|---|---|
| | $[QV]_{FI}$ | $[CS]_{FI}$ | $[EN]_{FI}$ | $[PT]_{FI}$ | $[T14]_{FI}$ | $[T15]_{FI}$ | $[SM]_{FI}$ | $[TV]_{FI}$ | $[QV]_{FI}$ | $[YH]_{FI}$ |
| *Image-LLMs: 8 uniformly sampled frames as inputs* | | | | | | | | | | |
| LLaVA-1.5 [58] | 3.0 | 9.2 | 1.9 | 7.6 | 7.7 | 8.0 | 3.1 | 1.7 | 33.6 | 28.2 |
| LLaVA-InternLM2 [11] | 0.2 | 5.2 | 0.1 | 0.4 | 0.2 | 0.1 | 0.1 | 0.3 | 24.4 | 40.1 |
| mPLUG-Owl2 [105] | 0.2 | 2.0 | 0.2 | 1.0 | 3.8 | 4.3 | 7.5 | 0.7 | 25.2 | 48.3 |
| XComposer [111] | 1.4 | 8.4 | 1.5 | 16.8 | 6.3 | 6.6 | 4.1 | 1.4 | 26.2 | 31.6 |
| Bunny-Llama3-V [31] | 10.2 | 3.8 | 0.1 | 2.6 | 6.3 | 6.4 | 0.3 | 0.4 | 20.0 | 41.2 |
| MiniCPM-V-2.5 [93] | 0.9 | 2.2 | 0.2 | 6.0 | 3.7 | 3.5 | 17.7 | 9.1 | 14.0 | 23.4 |
| Qwen-VL-Chat [6] | 6.8 | 25.8 | 4.0 | 14.2 | 9.2 | 8.6 | 25.0 | 7.7 | 25.6 | 43.2 |
| *Video-LLMs: each model's default numbers of frames as inputs* | | | | | | | | | | |
| Video-ChatGPT [66] | 2.8 | 11.1 | 1.3 | 24.0 | 10.2 | 11.0 | 12.9 | 4.0 | 25.4 | 32.2 |
| Video-LLaVA [53] | 3.0 | 11.1 | 1.9 | 23.9 | 10.2 | 11.0 | 0.0 | 0.6 | 26.2 | 31.6 |
| LLaMA-VID [51] | 1.4 | 9.6 | 1.2 | 14.5 | 4.6 | 5.0 | 0.4 | 2.4 | 25.8 | 34.2 |
| Video-LLaMA-2 [110] | 0.1 | 0.1 | 0.0 | 0.0 | 0.0 | 0.1 | 0.0 | 0.0 | 1.8 | 1.1 |
| PLLaVA [104] | 2.8 | 11.0 | 1.1 | 5.3 | 5.4 | 6.4 | 0.1 | 0.4 | 26.2 | 31.6 |
| VTimeLLM [33] | 2.8 | 12.3 | 1.9 | **27.7** | 13.9 | 13.0 | 15.0 | 16.8 | 26.2 | 31.6 |
| VTG-LLM [28] | 9.9 | 22.0 | 3.7 | 20.5 | 10.8 | 12.0 | 27.5 | 26.2 | 42.2 | 54.2 |
| TimeChat [82] | 15.1 | 37.2 | 3.8 | 11.1 | 10.8 | 8.4 | 30.6 | 27.7 | 33.2 | 47.7 |
| LITA [34] | 18.4 | 26.1 | 4.6 | 25.5 | 14.0 | 14.5 | **31.4** | **28.0** | 22.4 | 25.4 |
| **E.T. Chat** (Ours) | **26.9** | **50.4** | **10.2** | 22.9 | **34.4** | **35.0** | 26.8 | 24.1 | **69.4** | **55.7** |

Table 20: **Performance breakdown across source datasets on *dense captioning* and *complex understanding* tasks.** Abbreviations: [HI] HiREST, [YC] YouCook2, [CT] CrossTask, [HS] HT-Step, [PT] Perception Test, [QV] QVHighlights, [QE] QA-Ego4D.

| Method | DVC | | | | SLC | | | | TEM | | GVQ |
|---|---|---|---|---|---|---|---|---|---|---|---|
| | $[HI]_{FI}$ | $[HI]_{Sim}$ | $[YC]_{FI}$ | $[YC]_{Sim}$ | $[CT]_{FI}$ | $[CT]_{Sim}$ | $[HS]_{FI}$ | $[HS]_{Sim}$ | $[PT]_{Rec}$ | $[QV]_{Rec}$ | $[QE]_{Rec}$ |
| *Image-LLMs: 8 uniformly sampled frames as inputs* | | | | | | | | | | | |
| LLaVA-1.5 [58] | 20.6 | 12.2 | 8.3 | 10.9 | 0.6 | 10.1 | 1.3 | 8.9 | 13.9 | 1.6 | 0.0 |
| LLaVA-InternLM2 [11] | 16.4 | 9.4 | 17.5 | 7.6 | 0.1 | 5.1 | 0.0 | 4.3 | 13.0 | 1.3 | 1.5 |
| mPLUG-Owl2 [105] | 0.0 | 8.5 | 0.1 | 7.7 | 0.1 | 8.0 | 0.0 | 7.3 | 9.4 | 3.0 | 0.0 |
| XComposer [111] | 2.0 | 2.2 | 8.8 | 9.6 | 3.4 | 9.6 | 2.0 | 8.5 | 18.1 | 2.9 | 0.0 |
| Bunny-Llama3-V [31] | 11.6 | 8.8 | 15.4 | 8.9 | 0.1 | 7.4 | 0.0 | 7.7 | 11.8 | 2.6 | 0.0 |
| MiniCPM-V-2.5 [93] | 9.1 | 12.3 | 3.4 | 11.3 | 1.6 | 9.9 | 1.1 | 9.5 | 1.1 | 0.3 | 0.0 |
| Qwen-VL-Chat [6] | 14.9 | 12.2 | 19.8 | 15.4 | 9.5 | 11.7 | 3.0 | 14.4 | 4.2 | 2.3 | 1.5 |
| *Video-LLMs: each model's default numbers of frames as inputs* | | | | | | | | | | | |
| Video-ChatGPT [66] | 6.1 | 10.3 | 11.5 | 12.2 | 6.3 | 8.9 | 5.1 | 11.6 | 26.8 | 5.0 | 0.0 |
| Video-LLaVA [53] | 42.5 | 16.2 | 13.4 | 13.9 | 1.8 | 10.0 | 0.0 | 6.5 | 10.2 | 4.7 | 0.1 |
| LLaMA-VID [51] | 41.5 | 12.4 | 12.7 | 12.9 | 6.5 | 10.0 | 4.0 | 12.3 | 11.4 | 2.6 | 0.9 |
| Video-LLaMA-2 [110] | 1.2 | 9.2 | 0.1 | **19.8** | 0.0 | 14.4 | 0.1 | **16.1** | 0.0 | 0.0 | 0.1 |
| PLLaVA [104] | 5.6 | 10.0 | 21.0 | 11.1 | 9.3 | 10.1 | 10.2 | 13.4 | 6.9 | 1.3 | 1.2 |
| VTimeLLM [33] | 14.4 | 13.0 | 10.4 | 13.1 | 10.5 | 5.8 | 7.0 | 7.0 | 4.2 | 9.4 | 1.9 |
| VTG-LLM [28] | 45.4 | 19.3 | **35.0** | 18.0 | 20.3 | 14.1 | 21.3 | 14.7 | 14.1 | 3.7 | 1.4 |
| TimeChat [82] | 14.2 | 12.8 | 19.0 | 12.3 | 8.0 | 9.1 | 3.3 | 9.2 | 24.5 | 11.4 | 1.5 |
| LITA [34] | **47.0** | 15.9 | 32.4 | 18.5 | 21.3 | 12.1 | 20.6 | 12.3 | 20.3 | **11.7** | 2.2 |
| **E.T. Chat** (Ours) | 46.6 | **21.8** | 30.2 | 17.6 | **26.6** | **15.5** | **22.2** | 13.7 | **26.9** | 6.0 | **3.7** |

Table 21: **Performance under different IoU thresholds on** `[TVG]` **(left) and** `[EPM]` **(right).**

| Method | F1@0.1 | F1@0.3 | F1@0.5 | F1@0.7 | F1 |
|---|---|---|---|---|---|
| *Image-LLMs: 8 uniformly sampled frames as inputs* | | | | | |
| LLaVA-1.5 [58] | 19.1 | 4.7 | 0.5 | 0.0 | 6.1 |
| LLaVA-InternLM2 [11] | 7.4 | 3.0 | 0.4 | 0.0 | 2.7 |
| mPLUG-Owl2 [105] | 3.0 | 1.0 | 0.4 | 0.0 | 1.1 |
| XComposer [111] | 17.7 | 1.8 | 0.0 | 0.0 | 4.9 |
| Bunny-Llama3-V [31] | 24.4 | 3.2 | 0.5 | 0.0 | 7.0 |
| MiniCPM-V-2.5 [93] | 4.5 | 2.3 | 0.9 | 0.3 | 2.0 |
| Qwen-VL-Chat [6] | 31.4 | 19.1 | 10.4 | 4.1 | 16.2 |
| *Video-LLMs: each model's default numbers of frames as inputs* | | | | | |
| Video-ChatGPT [66] | 21.7 | 5.4 | 0.7 | 0.0 | 6.9 |
| Video-LLaVA [53] | 21.9 | 5.4 | 0.7 | 0.0 | 7.0 |
| LLaMA-VID [51] | 16.8 | 4.5 | 0.7 | 0.0 | 5.5 |
| Video-LLaMA-2 [110] | 0.3 | 0.0 | 0.0 | 0.0 | 0.1 |
| PLLaVA [104] | 21.4 | 5.4 | 0.7 | 0.0 | 6.9 |
| VTimeLLM [33] | 22.2 | 6.2 | 1.5 | 0.4 | 7.6 |
| VTG-LLM [28] | 38.8 | 16.1 | 6.6 | 2.2 | 15.9 |
| TimeChat [82] | 49.0 | 30.6 | 18.0 | 7.1 | 26.2 |
| LITA [34] | 50.9 | 25.0 | 8.8 | 4.3 | 22.2 |
| **E.T. Chat** (Ours) | **69.1** | **44.9** | **27.7** | **12.9** | **38.7** |

| Method | F1@0.1 | F1@0.3 | F1@0.5 | F1@0.7 | F1 |
|---|---|---|---|---|---|
| *Image-LLMs: 8 uniformly sampled frames as inputs* | | | | | |
| LLaVA-1.5 [58] | 5.8 | 1.2 | 0.4 | 0.2 | 1.9 |
| LLaVA-InternLM2 [11] | 0.2 | 0.0 | 0.0 | 0.0 | 0.1 |
| mPLUG-Owl2 [105] | 0.4 | 0.2 | 0.2 | 0.0 | 0.2 |
| XComposer [111] | 4.6 | 1.0 | 0.2 | 0.0 | 1.5 |
| Bunny-Llama3-V [31] | 0.2 | 0.0 | 0.0 | 0.0 | 0.1 |
| MiniCPM-V-2.5 [93] | 0.6 | 0.0 | 0.0 | 0.0 | 0.2 |
| Qwen-VL-Chat [6] | 8.4 | 5.2 | 1.6 | 0.8 | 4.0 |
| *Video-LLMs: each model's default numbers of frames as inputs* | | | | | |
| Video-ChatGPT [66] | 3.8 | 1.2 | 0.2 | 0.0 | 1.3 |
| Video-LLaVA [53] | 5.8 | 1.2 | 0.4 | 0.2 | 1.9 |
| LLaMA-VID [51] | 3.0 | 1.2 | 0.4 | 0.2 | 1.2 |
| Video-LLaMA-2 [110] | 0.0 | 0.0 | 0.0 | 0.0 | 0.0 |
| PLLaVA [104] | 3.2 | 1.0 | 0.2 | 0.0 | 1.1 |
| VTimeLLM [33] | 5.8 | 1.2 | 0.4 | 0.2 | 1.9 |
| VTG-LLM [28] | 9.2 | 4.6 | 0.6 | 0.4 | 3.7 |
| TimeChat [82] | 7.8 | 4.6 | 2.2 | 0.8 | 3.8 |
| LITA [34] | 12.6 | 4.4 | 1.0 | 0.4 | 4.6 |
| **E.T. Chat** (Ours) | **21.6** | **12.4** | **5.2** | **1.6** | **10.2** |

Table 22: **Performance under different IoU thresholds on** `[TAL]`.

| Method | F1@0.1 | F1@0.3 | F1@0.5 | F1@0.7 | F1 |
|---|---|---|---|---|---|
| *Image-LLMs: 8 uniformly sampled frames as inputs* | | | | | |
| LLaVA-1.5 [58] | 17.3 | 9.4 | 3.3 | 1.1 | 7.8 |
| LLaVA-InternLM2 [11] | 0.8 | 0.2 | 0.0 | 0.0 | 0.3 |
| mPLUG-Owl2 [105] | 6.7 | 3.8 | 1.3 | 0.4 | 3.0 |
| XComposer [111] | 22.0 | 12.2 | 4.5 | 0.9 | 9.9 |
| Bunny-Llama3-V [31] | 12.1 | 5.9 | 1.5 | 0.9 | 5.1 |
| MiniCPM-V-2.5 [93] | 9.6 | 5.0 | 2.3 | 0.7 | 4.4 |
| Qwen-VL-Chat [6] | 22.6 | 13.3 | 5.3 | 1.5 | 10.7 |
| *Video-LLMs: each model's default numbers of frames as inputs* | | | | | |
| Video-ChatGPT [66] | 32.6 | 18.7 | 6.8 | 2.1 | 15.1 |
| Video-LLaVA [53] | 32.6 | 18.8 | 6.8 | 2.0 | 15.0 |
| LLaMA-VID [51] | 17.8 | 10.0 | 3.5 | 0.8 | 8.0 |
| Video-LLaMA-2 [110] | 0.1 | 0.0 | 0.0 | 0.0 | 0.0 |
| PLLaVA [104] | 13.5 | 6.6 | 2.3 | 0.4 | 5.7 |
| VTimeLLM [33] | 39.3 | 21.9 | 9.0 | 2.7 | 18.2 |
| VTG-LLM [28] | 35.1 | 15.3 | 5.4 | 2.0 | 14.4 |
| TimeChat [82] | 24.8 | 10.2 | 4.0 | 1.4 | 10.1 |
| LITA [34] | 42.2 | 18.5 | 8.0 | 3.2 | 18.0 |
| **E.T. Chat** (Ours) | **59.2** | **33.3** | **20.2** | **10.3** | **30.8** |

Table 23: **Performance under more metrics on** `[DVC]`.

| Method | F1@0.1 | F1@0.3 | F1@0.5 | F1@0.7 | F1 | METEOR | Rouge-L | CIDEr | Sim |
|---|---|---|---|---|---|---|---|---|---|
| *Image-LLMs: 8 uniformly sampled frames as inputs* | | | | | | | | | |
| LLaVA-1.5 [58] | 29.6 | 17.0 | 7.9 | 3.3 | 14.5 | 0.9 | 1.8 | 2.1 | 11.5 |
| LLaVA-InternLM2 [11] | 36.1 | 20.4 | 8.3 | 3.0 | 16.9 | 1.0 | 1.6 | 1.6 | 8.5 |
| mPLUG-Owl2 [105] | 0.2 | 0.0 | 0.0 | 0.0 | 0.1 | 0.0 | 0.0 | 0.0 | 8.1 |
| XComposer [111] | 12.8 | 5.8 | 2.5 | 0.6 | 5.4 | 0.6 | 0.9 | 0.1 | 5.9 |
| Bunny-Llama3-V [31] | 32.1 | 15.2 | 5.1 | 1.8 | 13.5 | 0.9 | 1.6 | 2.2 | 8.8 |
| MiniCPM-V-2.5 [93] | 15.7 | 6.2 | 2.4 | 0.7 | 6.2 | 1.0 | 1.4 | 0.2 | 11.8 |
| Qwen-VL-Chat [6] | 39.6 | 20.4 | 6.9 | 2.6 | 17.4 | 1.3 | 2.2 | 2.7 | 13.8 |
| *Video-LLMs: each model's default numbers of frames as inputs* | | | | | | | | | |
| Video-ChatGPT [66] | 18.9 | 10.8 | 4.3 | 1.2 | 8.8 | 1.1 | 2.0 | 2.6 | 11.3 |
| Video-LLaVA [53] | 54.6 | 33.0 | 16.6 | 7.7 | 28.0 | 1.4 | 2.7 | 2.1 | 15.0 |
| LLaMA-VID [51] | 50.8 | 32.6 | 16.8 | 8.2 | 27.1 | 0.9 | 1.9 | 1.2 | 12.6 |
| Video-LLaMA-2 [110] | 1.2 | 0.7 | 0.6 | 0.0 | 0.6 | 0.0 | 0.0 | 0.0 | 14.5 |
| PLLaVA [104] | 29.3 | 15.9 | 6.1 | 2.0 | 13.3 | 1.3 | 2.4 | 3.7 | 10.6 |
| VTimeLLM [33] | 28.1 | 13.7 | 6.1 | 1.8 | 12.4 | 1.5 | 2.9 | 2.4 | 13.1 |
| VTG-LLM [28] | **81.0** | **51.6** | 22.0 | 6.2 | **40.2** | 2.8 | 5.2 | 8.5 | 18.6 |
| TimeChat [82] | 41.3 | 17.2 | 6.1 | 1.9 | 16.6 | 1.7 | 3.3 | 3.5 | 12.5 |
| LITA [34] | 78.5 | 49.2 | 23.5 | 7.6 | 39.7 | 3.3 | 5.2 | 7.6 | 17.2 |
| **E.T. Chat** (Ours) | 73.3 | 46.8 | **23.8** | 9.8 | 38.4 | 3.3 | **5.7** | **10.4** | **19.7** |

Table 24: **Performance under more metrics on** `[SLC]`.

| Method | F1@0.1 | F1@0.3 | F1@0.5 | F1@0.7 | F1 | METEOR | Rouge-L | CIDEr | Sim |
|---|---|---|---|---|---|---|---|---|---|
| *Image-LLMs: 8 uniformly sampled frames as inputs* | | | | | | | | | |
| LLaVA-1.5 [58] | 2.1 | 1.1 | 0.5 | 0.1 | 0.9 | 0.1 | 0.1 | 0.2 | 9.5 |
| LLaVA-InternLM2 [11] | 0.3 | 0.0 | 0.0 | 0.0 | 0.1 | 0.0 | 0.0 | 0.0 | 4.7 |
| mPLUG-Owl2 [105] | 0.1 | 0.1 | 0.0 | 0.0 | 0.1 | 0.0 | 0.0 | 0.0 | 7.7 |
| XComposer [111] | 5.9 | 3.4 | 1.1 | 0.4 | 2.7 | 0.3 | 0.3 | 0.0 | 9.0 |
| Bunny-Llama3-V [31] | 0.2 | 0.1 | 0.0 | 0.0 | 0.1 | 0.0 | 0.0 | 0.0 | 7.6 |
| MiniCPM-V-2.5 [93] | 4.3 | 0.8 | 0.2 | 0.2 | 1.4 | 0.2 | 0.2 | 0.1 | 9.7 |
| Qwen-VL-Chat [6] | 13.7 | 7.5 | 3.0 | 0.8 | 6.2 | 0.3 | 0.4 | 0.7 | 13.1 |
| *Video-LLMs: each model's default numbers of frames as inputs* | | | | | | | | | |
| Video-ChatGPT [66] | 12.2 | 7.1 | 2.5 | 0.9 | 5.7 | 0.4 | 0.7 | 1.2 | 10.2 |
| Video-LLaVA [53] | 1.8 | 1.1 | 0.5 | 0.2 | 0.9 | 0.0 | 0.0 | 0.1 | 8.3 |
| LLaMA-VID [51] | 14.0 | 4.6 | 1.7 | 0.6 | 5.2 | 0.2 | 0.3 | 0.3 | 11.1 |
| Video-LLaMA-2 [110] | 0.2 | 0.0 | 0.0 | 0.0 | 0.0 | 0.2 | 0.5 | 0.4 | **15.2** |
| PLLaVA [104] | 22.2 | 11.3 | 4.3 | 1.1 | 9.7 | 0.7 | 1.1 | 2.5 | 11.8 |
| VTimeLLM [33] | 19.1 | 10.4 | 4.1 | 1.4 | 8.7 | 0.4 | 0.6 | 0.9 | 6.4 |
| VTG-LLM [28] | **50.1** | 22.3 | 8.5 | 2.3 | 20.8 | 1.5 | 2.4 | 4.3 | 14.4 |
| TimeChat [82] | 15.9 | 4.7 | 1.5 | 0.4 | 5.6 | 0.6 | 1.0 | 1.2 | 9.2 |
| LITA [34] | 48.9 | 23.2 | 9.1 | 2.8 | 21.0 | 1.4 | 1.8 | 2.3 | 12.2 |
| **E.T. Chat** (Ours) | 45.8 | **28.8** | **15.8** | **7.2** | **24.4** | **2.4** | **3.2** | **6.2** | 14.6 |

Table 25: **Performance under different IoU thresholds on** `[TEM]` **(left) and** `[GVQ]` **(right).**

| Method | R@0.1 | R@0.3 | R@0.5 | R@0.7 | Rec | Method | R@0.1 | R@0.3 | R@0.5 | R@0.7 | Rec |
|---|---|---|---|---|---|---|---|---|---|---|---|
| *Image-LLMs: 8 uniformly sampled frames as inputs* | | | | | | *Image-LLMs: 8 uniformly sampled frames as inputs* | | | | | |
| LLaVA-1.5 [58] | 16.0 | 9.5 | 4.2 | 1.1 | 7.7 | LLaVA-1.5 [58] | 0.0 | 0.0 | 0.0 | 0.0 | 0.0 |
| LLaVA-InternLM2 [11] | 14.1 | 8.9 | 4.6 | 1.1 | 7.2 | LLaVA-InternLM2 [11] | 3.8 | 1.4 | 0.7 | 0.0 | 1.5 |
| mPLUG-Owl2 [105] | 12.4 | 7.3 | 3.8 | 1.1 | 6.2 | mPLUG-Owl2 [105] | 0.0 | 0.0 | 0.0 | 0.0 | 0.0 |
| XComposer [111] | 21.9 | 13.0 | 5.4 | 1.6 | 10.5 | XComposer [111] | 0.0 | 0.0 | 0.0 | 0.0 | 0.0 |
| Bunny-Llama3-V [31] | 13.8 | 9.1 | 4.5 | 1.3 | 7.2 | Bunny-Llama3-V [31] | 0.0 | 0.0 | 0.0 | 0.0 | 0.0 |
| MiniCPM-V-2.5 [93] | 1.5 | 0.7 | 0.5 | 0.2 | 0.7 | MiniCPM-V-2.5 [93] | 0.0 | 0.0 | 0.0 | 0.0 | 0.0 |
| Qwen-VL-Chat [6] | 7.7 | 3.5 | 1.3 | 0.4 | 3.2 | Qwen-VL-Chat [6] | 2.4 | 1.7 | 1.0 | 0.7 | 1.5 |
| *Video-LLMs: each model's default numbers of frames as inputs* | | | | | | *Video-LLMs: each model's default numbers of frames as inputs* | | | | | |
| Video-ChatGPT [66] | 32.1 | 18.9 | **9.6** | **2.7** | 15.9 | Video-ChatGPT [66] | 0.0 | 0.0 | 0.0 | 0.0 | 0.0 |
| Video-LLaVA [53] | 16.5 | 8.6 | 3.9 | 1.0 | 7.5 | Video-LLaVA [53] | 0.3 | 0.0 | 0.0 | 0.0 | 0.1 |
| LLaMA-VID [51] | 14.7 | 8.5 | 3.7 | 1.1 | 7.0 | LLaMA-VID [51] | 2.4 | 1.0 | 0.3 | 0.0 | 0.9 |
| Video-LLaMA-2 [110] | 0.0 | 0.0 | 0.0 | 0.0 | 0.0 | Video-LLaMA-2 [110] | 0.3 | 0.0 | 0.0 | 0.0 | 0.1 |
| PLLaVA [104] | 8.4 | 5.1 | 2.4 | 0.7 | 4.1 | PLLaVA [104] | 2.8 | 0.9 | 0.6 | 0.3 | 1.2 |
| VTimeLLM [33] | 16.1 | 7.5 | 2.7 | 0.9 | 6.8 | VTimeLLM [33] | 5.5 | 1.7 | 0.3 | 0.0 | 1.9 |
| VTG-LLM [28] | 17.4 | 10.9 | 5.2 | 2.1 | 8.9 | VTG-LLM [28] | 2.8 | 1.7 | 0.7 | 0.3 | 1.4 |
| TimeChat [82] | 38.6 | **21.7** | 8.9 | 2.7 | **18.0** | TimeChat [82] | 2.8 | 1.7 | 1.0 | 0.3 | 1.5 |
| LITA [34] | **40.4** | 15.8 | 6.2 | 1.8 | 16.0 | LITA [34] | 5.5 | 2.4 | 0.7 | 0.0 | 2.2 |
| **E.T. Chat** (Ours) | 36.9 | 20.2 | 6.7 | 2.0 | 16.5 | **E.T. Chat** (Ours) | **9.3** | **3.4** | **1.4** | **0.7** | **3.7** |

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
