# OpenReview forum: "E.T. Bench: Towards Open-Ended Event-Level Video-Language Understanding"
_NeurIPS.cc/2024/Datasets_and_Benchmarks_Track — NeurIPS 2024 Track Datasets and Benchmarks Poster_

### Official Review · Reviewer_yb1y · 2024-07-17
**This paper proposes an Event-Level Video Understanding Benchmark dataset, a video-LLM baseline model, and a instruction-tuning dataset.**

**Rating:** 8
**Confidence:** 3
**Correctness:** Yes.
**Clarity:** Yes, with lots of detailed information.

**Review:**

Although the data are collected and combined from other public dataset, the E.T. Bench is still highly original from my point of view, featuring a unique focus on event-level assessment with careful filtering, which sets it apart from existing benchmarks. Both E. T. Bench and the E.T. 164K are in good quality based on the experiment results. E. T. Bench distinguishes the performance of different models, and  E. T. 164K ensures the good performance of the E. T. Chat model.

Pros
1. The paper is well-structured, providing a clear introduction to the motivation behind the benchmark and a detailed description of the benchmark's building.
2. Comprehensive evaluation of various models provides a robust validation of the benchmark's effectiveness and the associated challenges in video-language understanding.
3. A new video-LLM baseline model (E.T. Chat) is also proposed.

Cons
1. It might be challenging to use this dataset to accurately rank the general capabilities of Video-LLMs because the models are evaluated on multiple subsets separately, lacking a unified metric or comprehensive data mixture to assess overall performance across diverse scenarios.
2. Other baseline video models may not be tuned on the similar defined tasks, which may leads to an unfair comparison. It might be better to report the results after being tuned on the E. T. 164K.
3. Due to the benchmark are built via public datasets, some models (especially the commercial models) could already be trained on those public datasets, which leads to an unfair comparison.

**Strengths:**

See Pros in Review

**Additional Feedback:**

I hope the benchmark and the E.T. Chat model can be open-sourced in the future.

**Documentation:**

Not sure. There's no link to access the dataset right now. The json files of all subsets are provided. The benchmark seems to be reproducible.

**Ethics:**

No.

**Limitations:**

Yes, I see that the authors have admitted the limitation of the possible data leakage and the low-resolution problem of the model.

**Opportunities For Improvement:**

See Cons in Review

**Relation To Prior Work:**

Yes, with a table of quantitative comparison.

**Summary And Contributions:**

This paper introduces a comprehensive video understanding benchmark E.T. Bench to evaluate Video-LLMs. This benchmark includes multiple tasks such as referring, grounding, dense captioning, and complex understanding, focusing on event-level and time sensitive understanding. Additionally, the authors propose a new baseline model, E.T. Chat, and a instruct-tuning dataset, E.T. 164K. Comprehensive analyses and experiments are conducted, with rich information of the benchmark collection, model comparison, and other details.

---

> ### Author Rebuttal · Authors · 2024-08-17
>
> > Q1: Lacking a unified metric to accurately rank the general capabilities of Video-LLMs.
>
> Thanks for your insight. We agree that designing a unified metric to cover all scenarios would be useful for ranking. However, E.T. Bench differs from existing MCQ-based benchmarks in terms of diverse task formulations, yielding heterogeneous evaluation metrics for different tasks. We have tried our best to unify and reduce them to 5 metrics for 12 tasks. Such unification was discussed at L166 - L171 in the supplementary material, and we illustrate the 5 metrics below.
>
> 1. Acc$_{ref}$: Averaged accuracy on referring tasks (RAR, ECA, RVQ);
> 2. F1$_{gnd}$: Averaged F1 score on grounding tasks (TVG, EPM, TAL, EVS, VHD);
> 3. F1$_{cap}$: Averaged F1 score on dense captioning tasks (DVC, SLC);
> 4. Sim$_{cap}$: Averaged sentence similarity score on dense captioning tasks (DVC, SLC);
> 5. Rec$_{com}$: Averaged recall on complex understanding tasks (TEM, GVQ);
>
> On the other hand, as tasks in E.T. Bench are diverse and have different input-output formats, users can diagnose their models’ capabilities under multiple scenarios and choose the metrics that best align with their applications.
>
> > Q2: Better to report the results of other models after being tuned on E.T. 164K.
>
> We have conducted an ablation study to disentangle the effect of model and instruction-tuning data designs at L218 - L223 in the supplementary material. The results are presented below. We chose a representative conventional Video-LLM (LLaMA-VID [1]) and a strong time-sensitive Video-LLM (TimeChat [2]) as our baselines. Their performances were evaluated after training on different instruction-tuning datasets.
>
> | ID | Model | IT Dataset | Acc$_{ref}$ | F1$_{gnd}$ | F1$_{cap}$ | Sim$_{cap}$ | Rec$_{com}$ |
> | :-: |:-: |:-: |:-: |:-: |:-: |:-: |:-: |
> | A | LLaMA-VID [1] | 723K Corpus [1] | 32.5 | 8.9 | 16.4 | 11.9 | 4.0 |
> | B | LLaMA-VID [1] | E.T. 164K (Ours) | 31.3 | 16.1 | 20.3 | 12.0 | 7.8 |
> | C | TimeChat [2] | TimeIT [2] | 27.7 | 22.0 | 21.6 | 10.9 | 9.8 |
> | D | TimeChat [2] | E.T. 164K (Ours) | 29.5 | 24.6 | 22.4 | 12.1 | 11.7 |
> | E | E.T. Chat (Ours) | TimeIT [2] | 34.9 | 22.4 | 20.2 | 13.5 | 6.9 |
> | F | E.T. Chat (Ours) | E.T. 164K (Ours) | **36.7** | **30.5** | **27.5** | **13.9** | **13.3** |
>
> When trained on the same E.T. 164K dataset (comparing B, D, and F), E.T. Chat achieves the best performance across all capabilities, demonstrating the effectiveness of the architectural design. When compared with existing instruction-tuning datasets (comparing A & B, C & D, and E & F), our E.T. 164K can also steadily yield stronger performance on different models, proving the significance of the instruction-tuning data. Results of more models after being tuned on E.T. 164K will be included in our revision.
>
> > Q3: Potential data leakage on some models (especially commercial ones).
>
> Thanks for pointing it out. During the development of E.T. Bench, we carefully selected the data sources that are not used for training in mainstream instruction-tuning datasets, and we only collected the videos and annotations from the test (if available) or validation splits. However, for commercial models, we agree that it’s hard to guarantee all the videos and annotations in E.T. Bench were excluded from their training. Hence, the evaluation results of commercial MLLMs might be inevitably slightly unfair. This is a common issue in Video-LLM benchmarks [3, 4, 5], and we hope that this could be alleviated by maintaining a private & self-annotated test set in the future.
>
> > [Documentation] Seems to be reproducible but only the JSON files are available right now.
>
> All the necessary materials for the benchmark (videos, annotations files, and evaluation scripts) will be publicly available to ensure reproducibility.
>
> > [Additional Feedback] I hope the benchmark and model can be open-sourced in the future.
>
> Yes, all the claimed contributions in this paper, including the E.T. Bench (videos, annotations, evaluation script), the E.T. Chat model (code, checkpoints, documentation), and the E.T. 164K instruction-tuning dataset (videos, annotations, pre-processing code) will be open-sourced. We also encourage the community to pay more attention to this area and develop stronger time-sensitive MLLMs with reference to our benchmark.
>
> We hope that the responses above can address your concerns. More discussions are welcomed if you have any further questions. Thank you again for your valuable review!
>
> [1] LLaMA-VID: An Image is Worth 2 Tokens in Large Language Models. Li et al., ECCV 2024.\
> [2] TimeChat: A Time-sensitive Multimodal Large Language Model for Long Video Understanding. Ren et al., CVPR 2024.\
> [3] MVBench: A Comprehensive Multi-modal Video Understanding Benchmark. Li et al., CVPR 2024.\
> [4] Video-MME: The First-Ever Comprehensive Evaluation Benchmark of Multi-modal LLMs in Video Analysis. Fu et al., arXiv 2024.\
> [5] TempCompass: Do Video LLMs Really Understand Videos? Liu et al., arXiv 2024.

---

> > ### Comment · Reviewer_yb1y · 2024-08-21
> > **Response to Rebuttal**
> >
> > I appreciate your effort and I will keep my score.

---

> ### Author Response · Authors · 2024-08-22
>
> We are grateful for your active involvement in reviewing our work and thank you for the positive feedback! We will keep maintaining our benchmark, model, and IT dataset to facilitate future research in this direction.

---

### Official Review · Reviewer_ct4u · 2024-07-22
**Good dataset contribution maybe overclaiming on benchmarking and need for new baseline.**

**Rating:** 6
**Confidence:** 4
**Correctness:** Yes
**Clarity:** It is well enough, but please proof r…

**Review:**

I like the dataset being proposed, even though it is just a mesh-up of existing datasets. Having an overarching structure specifically for event-level understanding tasks that contains these data is still a good contribution.

On the other hand, I think the text is overclaiming a bit for image- and video-LLMs are underperforming the task. Specifically, for the main results in Table 3, it is clear that commercial LLMs are still outperforming the proposed E.T. Chat model even after instruction tuning. Is this gap expected?

Given these, I am leaning on the slight positive side as I think it has a good contribution, but maybe the claims need to be toned down.

**Strengths:**

- A structural event-level video understanding dataset
- Contains a large number of tasks, long video, as well as even-level annotations.

**Additional Feedback:**

No additional feedback.

**Documentation:**

Looks good.

**Ethics:**

No.

**Limitations:**

I think the limitation is adequately described.

**Opportunities For Improvement:**

- The writing needs to improve. I found quite a few typos throughout the manuscript, for example, "bagpack".
- Not sure how strong is the proposed E.T. Chat and the value of the instruction tuning dataset that it is trained on.

**Relation To Prior Work:**

Yes

**Summary And Contributions:**

This paper introduces a dataset for fine-grain event-level assessment. It contains 7.8 samples, 12 tasks with 7.7 videos. Image and video LLMs were evaluated on the dataset, and they show the video-level understanding capability does not generalize to event levels. The author then proposed a new baseline method with a tuning dataset to address this.

---

> ### Author Rebuttal · Authors · 2024-08-17
>
> Many thanks for your constructive comments! We are encouraged by your recognition on our contributions. Below we provide responses to your concerns in detail.
>
> > Q1: Is it expected that commercial MLLMs still outperform E.T. Chat with instruction-tuning?
>
> We believe this is expected as commercial LLMs generally have much larger scales and were trained with much more data. Moreover, as discussed in limitations and also pointed out by reviewer yb1y, we could not guarantee that all the videos and annotations in E.T. Bench were excluded from training these models. Hence the comparison between commercial LLMs and E.T. Chat might be inevitably slightly unfair. This is a common issue in Video-LLM benchmarks [1, 2, 3] and we hope that this could be alleviated by maintaining a private & self-annotated test set in the future.
>
> > Q2: I think the submission has a good contribution, but maybe the claims need to be toned down.
>
> Thank you for the suggestion! Our claim that Image- and Video-LLMs are underperforming the task is based on the evaluation results of open-source models. We will carefully revise these statements to ensure no overclaiming and misunderstanding.
>
> > Q3: Typos were found in the manuscript.
>
> Thanks for pointing them out and we apologize for the mistakes. A careful grammar check has been conducted and all the typos will be corrected in the next version.
>
> > Q4: Not sure how strong is the proposed E.T. Chat and the value of E.T. 164K.
>
> We provided an ablation study on the effectiveness of E.T. Chat and E.T. 164K at L218 - L223 in the supplementary material. The results are presented below. We chose a representative conventional Video-LLM (LLaMA-VID [4]) and a strong time-sensitive Video-LLM (TimeChat [5]) as our baselines. Their performances were evaluated after training on different instruction-tuning datasets.
>
> | ID | Model | IT Dataset | Acc$_{ref}$ | F1$_{gnd}$ | F1$_{cap}$ | Sim$_{cap}$ | Rec$_{com}$ |
> | :-: |:-: |:-: |:-: |:-: |:-: |:-: |:-: |
> | A | LLaMA-VID [4] | 723K Corpus [4] | 32.5 | 8.9 | 16.4 | 11.9 | 4.0 |
> | B | LLaMA-VID [4] | E.T. 164K (Ours) | 31.3 | 16.1 | 20.3 | 12.0 | 7.8 |
> | C | TimeChat [5] | TimeIT [5] | 27.7 | 22.0 | 21.6 | 10.9 | 9.8 |
> | D | TimeChat [5] | E.T. 164K (Ours) | 29.5 | 24.6 | 22.4 | 12.1 | 11.7 |
> | E | E.T. Chat (Ours) | TimeIT [5] | 34.9 | 22.4 | 20.2 | 13.5 | 6.9 |
> | F | E.T. Chat (Ours) | E.T. 164K (Ours) | **36.7** | **30.5** | **27.5** | **13.9** | **13.3** |
>
> When trained on the same E.T. 164K dataset (comparing B, D, and F), E.T. Chat achieves the best performance across all capabilities, demonstrating the effectiveness of the architectural design. When compared with existing instruction-tuning datasets (comparing A & B, C & D, and E & F), our E.T. 164K can also steadily yield stronger performance on different models, proving the significance of the instruction-tuning data.
>
> We hope that the responses above can address your concerns. More discussions are welcomed if you have any further questions. Thank you again for your valuable feedback!
>
> [1] MVBench: A Comprehensive Multi-modal Video Understanding Benchmark. Li et al., CVPR 2024.\
> [2] Video-MME: The First-Ever Comprehensive Evaluation Benchmark of Multi-modal LLMs in Video Analysis. Fu et al., arXiv 2024.\
> [3] TempCompass: Do Video LLMs Really Understand Videos? Liu et al., arXiv 2024.\
> [4] LLaMA-VID: An Image is Worth 2 Tokens in Large Language Models. Li et al., ECCV 2024.\
> [5] TimeChat: A Time-sensitive Multimodal Large Language Model for Long Video Understanding. Ren et al., CVPR 2024.

---

> > ### Author Response · Authors · 2024-08-27
> >
> > Dear Reviewer ct4u,
> >
> > We are wholeheartedly grateful for your time dedicated to improving our work and for the constructive feedback.
> >
> > The end of author-reviewer discussion phase is approaching, and we hope that our rebuttal has addressed your concerns. Please kindly let us know if you have any further questions or suggestions. We would greatly appreciate any further feedback or confirmation of our responses. Thank you!
> >
> > Sincerely,
> >
> > Authors of Paper 1536

---

### Official Review · Reviewer_T7B9 · 2024-07-24
**The paper highlights the limitations of existing video-language understanding benchmarks, which primarily focus on video-level question-answering and lack fine-grained event-level assessment.**

**Rating:** 6
**Confidence:** 4
**Correctness:** Yes
**Clarity:** Yes

**Review:**

Pros
1. E.T. Bench provides a comprehensive evaluation framework for assessing the event-level and time-sensitive understanding capabilities of Video-LLMs. Along with the bench, the 3-level task taxonomy is well-structured and clearly defines the capabilities and tasks being evaluated. This facilitates a more nuanced understanding of model performance and helps identify specific areas for improvement.
2. The novel timestamp processing design and the multi-event instruction-tuning data effectively address the limitations of existing models and datasets.
3. The authors conduct extensive evaluations on a wide range of models, including open-source Image-LLMs, Video-LLMs, and commercial MLLMs. This provides a comprehensive understanding of the current state-of-the-art and highlights the challenges and opportunities in the field.

Cons
1. The benchmark could be expanded to include additional tasks that further probe the capabilities of Video-LLMs. For example, tasks involving multi-modal reasoning, causal inference, or temporal logic could be valuable additions.
2. The paper does not delve deeply into analyzing the underlying reasons for model successes and failures.
3. While the paper mentions other time-sensitive Video-LLMs, it does not provide a direct comparison with them. This would strengthen the paper's argument for the effectiveness of E.T. Chat and provide a more comprehensive evaluation of the state-of-the-art.

**Strengths:**

1. E.T. Bench provides a comprehensive evaluation framework for assessing the event-level and time-sensitive understanding capabilities of Video-LLMs. Along with the bench, the 3-level task taxonomy is well-structured and clearly defines the capabilities and tasks being evaluated. This facilitates a more nuanced understanding of model performance and helps identify specific areas for improvement.
2. The novel timestamp processing design and the multi-event instruction-tuning data effectively address the limitations of existing models and datasets.
3. The authors conduct extensive evaluations on a wide range of models, including open-source Image-LLMs, Video-LLMs, and commercial MLLMs. This provides a comprehensive understanding of the current state-of-the-art and highlights the challenges and opportunities in the field.

**Additional Feedback:**

Please refer to the weaknesses and questiones part.

**Documentation:**

Yes

**Limitations:**

1. The benchmark relies on existing datasets, which may contain biases that could influence model performance. The authors should acknowledge these potential biases and discuss their impact on the evaluation results.

**Opportunities For Improvement:**

1. How can the benchmark be further expanded to include more diverse and challenging tasks?
2. What are the limitations of the proposed timestamp processing design, and how can they be addressed?

**Relation To Prior Work:**

Yes

**Summary And Contributions:**

The paper highlights the limitations of existing video-language understanding benchmarks, focusing on video-level question-answering and lack fine-grained event-level assessment. To address this, the authors introduce E.T. Bench, a benchmark encompassing 12 tasks categorized within a 3-level task taxonomy. These tasks evaluate four capabilities for time-sensitive video understanding: referring, grounding, dense captioning, and complex understanding.
The authors demonstrate that state-of-the-art models struggle on E.T. Bench, particularly on tasks requiring fine-grained temporal information. They attribute this to limitations in current model design and training data, specifically the discrete next-token prediction paradigm and the lack of multi-event training data. To address these issues, they propose E.T. Chat, a novel Video-LLM that reformulates timestamp prediction as an embedding matching problem, and E.T. 164K, an instruction-tuning dataset tailored for multi-event and time-sensitive scenarios. Extensive evaluations demonstrate the effectiveness of their proposed model and dataset.

---

> ### Author Rebuttal · Authors · 2024-08-17
>
> We sincerely thank you for the detailed and constructive comments. Below we provide point-by-point responses to address your concerns.
>
> > Q1: The benchmark can be further expanded to include more diverse and challenging tasks.
>
> Thank you for the insightful suggestion. We agree that these are essential capabilities for general-purpose Video-LLMs, and they have been widely studied in video-level understanding benchmarks but not under event-level and time-sensitive scenarios. We take this insight as part of our roadmap towards expanding E.T. Bench to more diverse and challenging scenarios.
>
> Specifically, the current version of E.T. Bench primarily focuses on closing the gap between video-level holistic perception and event-level & time-sensitive understanding. Therefore, although it contains two complex understanding tasks (TEM and GVQ), this benchmark is designed to provide comprehensive evaluations specially for event awareness and timestamp processing abilities. Even on these relatively “simple” tasks (that rely more on perception than reasoning), the evaluation results can still reveal existing MLLMs’ limitations on timestamp processing, which would motivate the community to pay more attention to such scenarios.
>
> As for our future work, the next generation of E.T. Bench will focus more on complex scenarios that require world-knowledge and strong reasoning capabilities, while preserving its nature of event awareness and time-sensitivity. Taking the mentioned “causal inference” for example, such a task could be formulated as asking the model “Why is the pancake undercooked in some areas and overcooked in others?”, and the correct response would be “Because at around 32s, the chef forgot to preheat the oven before cooking”. Answering such a question requires both 1) the knowledge about cooking and 2) the ability of time-sensitive causal reasoning. Apart from this, we are also exploring to blend these tasks with more flexible and natural conversation styles to align them with real-world applications.
>
> > Q2: Analyzing the underlying reasons for model successes and failures.
>
> Thank you for pointing this out. Through the evaluations, we observe the following key aspects that may influence model successes and failures:
>
> 1. **Instruction-following ability.** Tasks in E.T. Bench are diverse in terms of input-output formats, thus the models are required to exhibit good instruction-following abilities to generate correct responses. We observe that some Video-LLMs, e.g., Video-LLaVA and Video-LLaMA-2, struggle to generate responses with desired formats even with carefully designed prompts.
> 2. **Time awareness.** Some models, e.g., Video-ChatGPT, PLLaVA, and most Image-LLMs, can successfully follow user instructions, but fail to understand and/or predict timestamps. This is due to the lack of time awareness caused by the aggressive temporal pooling layers and missing event-level data for training.
> 3. **Continuous numerical processing ability.** Some time-sensitive Video-LLMs (e.g., TimeChat) have good instruction-following abilities and time awareness, but are still sub-optimal as they cannot model the continuous characteristics of timestamps, leading to many incorrect time predictions that are far from the ground truths.
>
> > Q3: Lack of direct comparison with other time-sensitive Video-LLMs.
>
> We clarify that comparisons between E.T. Chat and existing representative time-sensitive Video-LLMs, including VTimeLLM, TimeChat, and LITA, are presented in both the main paper (Table 3) and the supplementary material (Table 18 - 22). Another mentioned model, i.e., Momentor, was not evaluated because the authors did not release their checkpoint. Please refer to the general rebuttal for a summary of comparisons. It shows that our model outperforms other time-sensitive Video-LLMs on most tasks, demonstrating the effectiveness of the proposed method.
>
> > Q4: What are the limitations of the proposed timestamp processing design, and how can they be addressed?
>
> The proposed “timestamp generation as embedding matching” design in E.T. Chat can be regarded as a trade-off between *discrete next-token prediction* and *continuous numerical processing*. To implement this design, a new special token (<vid> token) should be added to the LLM’s vocabulary. Although this is much simpler than existing methods that leverage multiple (usually around 100) special tokens to represent quantized timestamps, adding new tokens to a pre-trained LLM is still not encouraged as it would affect the learned distributions.
>
> We propose three potential solutions to address this limitation.
>
> 1. **Carefully design the instruction-tuning data by mixing samples with and without <vid> tokens.** With an appropriate data mixing strategy, the model would implicitly learn when to generate <vid> token and when not to.
> 2. **Use explicit instructions to trigger <vid> prediction.** Through this way, the user can actively control the model behaviors by writing different instructions.
> 3. **Explore other designs that do not require special tokens.** A straightforward idea would be regressing timestamps directly from the features of text tokens. One might consider using a binary classifier to determine whether to regress timestamps on each token, and leverage an MLP to do regression.
>
> > Q5: Authors should acknowledge the potential dataset biases and discuss their impact on evaluation results.
>
> We agree that some existing datasets inevitably contain biases. However, this problem shall be alleviated by collecting videos and annotations from multiple sources (and multiple domains) for each task. Moreover, the evaluations on E.T. Bench are performed under zero-shot settings, such that the results would not be influenced by overfitting their training sets. We will provide more discussions on such phenomenon in our revision.
>
> We hope that the responses above can address your concerns. More discussions are welcomed if you have any further questions. Thank you!

---

> > ### Author Response · Authors · 2024-08-27
> >
> > Dear Reviewer T7B9,
> >
> > We are wholeheartedly grateful for your time dedicated to improving our work and for the constructive feedback.
> >
> > The end of author-reviewer discussion phase is approaching, and we hope that our rebuttal has addressed your concerns. Please kindly let us know if you have any further questions or suggestions. We would greatly appreciate any further feedback or confirmation of our responses. Thank you!
> >
> > Sincerely,
> >
> > Authors of Paper 1536

---

### Official Review · Reviewer_77Mr · 2024-07-24
**E.T.Bench**

**Rating:** 8
**Confidence:** 5
**Correctness:** The paper is well-stated.
**Clarity:** The paper is clearly written and easy…

**Review:**

**Pros**
1. in general, the dataset scale and variety are highlighted; E.T.Bench considers the most practical cases in video understanding.
2. baseline model E.T.164K is solid in its design, instruction training, and also model performance, which contributes to the completeness of the paper.
3. ablation studies on the model design choice of E.T.Chat is solid.

**Cons**
1. The figure presentation can still be improved; please refer to **Opportunities For Improvement**

**Strengths:**

1. E.T.Bench considers a well-defined taxonomy of 4 capabilities and a large variety of 12 fine-grained video understanding tasks. Besides, the instruction-tuning dataset is also very valuable. High-quality datasets are always a cornerstone of video understanding research.
2. E.T. 164K also presents a strong baseline for future video understanding research compared to other strong baselines on E.T.Bench.
3. the paper also has provided solid insights based on the experimental results.

**Additional Feedback:**

Generally, the paper is methodical and comprehensive. I would suggest acceptance.

**Documentation:**

-[x] data collection and organization
-[x] availability and maintenance
-[x] ethical and responsible use

**Limitations:**

Yes

**Opportunities For Improvement:**

I only have some minor suggestions for the presentation:
1. In Fig. 3 Left, statistics of verbs and nouns in the dataset could be shown in a more straightforward and informative way, like a histogram, than a word cloud, in my opinion.
2. In Fig. 4, it would be clearer if notations like $E_{att}$, $E_p$, $E_q$, $\mathbf Q_t$, and $\mathbf P_t$ are also illustrated for references.

**Relation To Prior Work:**

Yes, the paper especially considers more fine-grained level video understanding tasks compared to most existing benchmarks and integrates multiple tasks of different formats in a uniform framework (e.g., grounding and qa).

**Summary And Contributions:**

The paper proposes a new video understanding dataset, E.T.Bench, that covers a variety of fine-grained tasks and also a strong baseline model, E.T.Chat, and a new instruction-tuning dataset for pre-training. The paper also benchmarks a range of important existing models and shares important insights into SOTA models' video understanding ability.

---

> ### Author Rebuttal · Authors · 2024-08-16
>
> Thank you for the constructive feedback! We are greatly encouraged by your positive review. Below we provide detailed responses to your concerns and suggestions.
>
> > Q1: In Fig. 3 (left), statistics of verbs and nouns could be shown by histograms.
>
> Thanks for your suggestion. We plot the frequency distributions of verbs and nouns in E.T. Bench via histograms. Please see the attached PDF (Fig. A and Fig. B) for details. Note that the queries in our benchmark encompass 461 different verbs and 3090 different nouns in total, while we only visualize the top 25 words for clarity. The distribution histograms could indeed better demonstrate the diversity of queries in E.T. Bench.
>
> > Q2: It would be clearer to include notations like E$_{att}$, E$_p$, E$_q$, Q$_t$, and P$_t$ in Fig. 4.
>
> To better illustrate these notations, we provide a new figure to supplement Fig. 4 and clarify the detailed procedure in the frame compressor. Please refer to the attached PDF (Fig. C) for details. The notations and data flow shall strictly follow the descriptions in Section 3.1 (Frame Compression). With this new figure, the illustration for the frame compression module would be much clearer.
>
> All these new figures will be included in our revision to ensure clearness and good readability. More discussions are welcomed if you have any further questions. Thank you again for your valuable suggestions!

---

> > ### Comment · Reviewer_77Mr · 2024-08-22
> >
> > Thanks for the response. I will keep my ratings.

---

> > > ### Author Response · Authors · 2024-08-22
> > >
> > > We are glad that you recognize the value of our work. Thank you for your constructive feedback!

---

### Author Rebuttal · Authors · 2024-08-17

We sincerely thank all the reviewers for their time and valuable feedback. We are encouraged by the reviewers' recognition that:

- Our motivation is clear and significant - E.T. Bench considers the most practical cases in video understanding. (77Mr, yb1y)
- The benchmark is well-structured with clearly defined capabilities and tasks. (77Mr, T7B9, ct4u, yb1y)
- The evaluations are comprehensive and can highlight the challenges and opportunities in the field. (77Mr, T7B9, yb1y)
- E.T. Chat model is solid in terms of the novel timestamp processing design. (77Mr, T7B9, yb1y)
- E.T. 164K dataset is valuable, with good quality, and can effectively address existing limitations. (77Mr, T7B9, yb1y)

In this rebuttal, we update the benchmark by adding two evaluations:

1. VTG-LLM [1]: a recent open-source time-sensitive Video-LLM focusing on grounding-related applications.
2. E.T. Chat*: A newly trained version that is aligned and instruction-tuned from scratch instead of being initialized from LLaMA-VID [2] checkpoints as in the original paper, which is more reasonable and brings slight performance gains in most tasks.

The updated comparisons among time-sensitive Video-LLMs are summarized below.

| Method | LLM Size | RAR$_{Acc}$ | EVC$_{Acc}$ | RVQ$_{Acc}$| TVG$_{F1}$ | EPM$_{F1}$ | TAL$_{F1}$ | EVS$_{F1}$ | VHD$_{F1}$ | DVC$_{F1}$ | DVC$_{Sim}$ | SLC$_{F1}$ | SLC$_{Sim}$ | TEM$_{Rec}$ | GVQ$_{Rec}$ |
|-|-|-|-|-|-|-|-|-|-|-|-|-|-|-|-|
| VTimeLLM [3] | 7B | 28.4 | 31.0 | 29.2 | 9.3 | 1.9 | 18.2 | 15.9 | 28.9 | 22.3 | 13.2 | 8.8 | 6.3 | 6.8 | 1.9 |
| TimeChat [4] | 7B | 30.8 | 27.6 | 24.6 | 26.6 | 3.9 | 10.1 | 29.1 | 40.5 | 30.9 | 12.5 | 12.3 | 9.0 | 18.0 | 1.5 |
| VTG-LLM [1] | 7B | 6.6 | 12.0 | 7.8 | 17.0 | 3.7 | 14.4 | 26.8 | 48.2 | 39.5 | 18.6 | 20.8 | 14.3 | 8.9 | 1.4 |
| LITA [5] | 13B | 33.0 | **40.8** | 27.2 | 21.0 | 4.6 | 18.0 | **29.7** | 23.9 | 39.7 | 17.1 | 21.0 | 12.1 | 16.0 | 2.2 |
| E.T. Chat* (Ours) | 7B | **36.8** | 35.6 | **38.4** | **34.7** | **7.0** | **28.7** | 29.2 | **61.8** | **39.8** | **19.0** | **25.1** | **14.5** | **20.7** | **4.3** |

Please refer to the comments under your reviews for detailed responses. Thank you!

[1] VTG-LLM: Integrating Timestamp Knowledge into Video LLMs for Enhanced Video Temporal Grounding. Guo et al., arXiv 2024.\
[2] LLaMA-VID: An Image is Worth 2 Tokens in Large Language Models. Li et al., ECCV 2024.\
[3] VTimeLLM: Empower LLM to Grasp Video Moments. Huang et al., CVPR 2024.\
[4] TimeChat: A Time-sensitive Multimodal Large Language Model for Long Video Understanding. Ren et al., CVPR 2024.\
[5] LITA: Language Instructed Temporal-Localization Assistant. Huang et al., ECCV 2024.

---

### Decision · Program_Chairs · 2024-09-26

**Decision:**

Accept (Poster)

**Comment:**

This paper contributes a comprehensive benchmark for fine-grain event-level assessment (E.T. Bench), which includes multiple tasks such as referring, grounding, dense captioning, and complex understanding.In addition, this paper introduces a new baseline model (E.T. Chat) and a instruct-tuning dataset (E.T. 164K). Thus paper also benchmarks a range of important existing models and shares important insights into SOTA models' video understanding ability.

All reviewers lean towards accepting this paper and find the following strengths:

- A new video understanding dataset (E.T.Bench) with a variety of fine-grained tasks and large scale
- Clear introduction and motivation behind the benchmark
- Extensive evaluations on a wide range of models and a new introduced video-LLM baseline (E.T. Chat)